# Towards Efficient and Effective Adversarial Training

**Gaurang Sriramanan**[*], **Sravanti Addepalli**[*], **Arya Baburaj**, **R.Venkatesh Babu**
Video Analytics Lab, Department of Computational and Data Sciences
Indian Institute of Science, Bangalore, India

## Abstract

The vulnerability of Deep Neural Networks to adversarial attacks has spurred immense interest towards improving their robustness. However, present state-of-the-art adversarial defenses involve the use of 10-step adversaries during training, which renders them computationally infeasible for application to large-scale datasets. While the recent single-step defenses show promising direction, their robustness is not on par with multi-step training methods. In this work, we bridge this performance gap by introducing a novel Nuclear-Norm regularizer on network predictions to enforce function smoothing in the vicinity of data samples. While prior works consider each data sample independently, the proposed regularizer uses the joint statistics of adversarial samples across a training minibatch to enhance optimization during both attack generation and training, obtaining state-of-the-art results amongst efficient defenses. We achieve further gains by incorporating exponential averaging of network weights over training iterations. We finally introduce a Hybrid training approach that combines the effectiveness of a two-step variant of the proposed defense with the efficiency of a single-step defense. We demonstrate superior results when compared to multi-step defenses such as TRADES and PGD-AT as well, at a significantly lower computational cost.

## 1  Introduction

The past decade has witnessed the rise of Deep Learning based systems and solutions in many applications related to Computer Vision, Natural Language Processing and Speech Processing. Despite the remarkable success of Deep Networks, they are known to be susceptible to crafted imperceptible noise called Adversarial Attacks [27], which can have disastrous implications when deployed in critical applications such as self-driving cars, medical diagnosis and surveillance systems. The robustness of Deep Networks to Adversarial Attacks has been of prime importance, since this relates to improving their worst-case performance [6].

Early attempts of improving robustness to adversarial attacks could be broadly classified into input pre-processing based defenses [4, 32, 25, 13] and adversarial training methods [11, 20]. While the input pre-processing based defenses were computationally cheap, they primarily involved masking of gradients to prevent the generation of strong attacks. Such methods were later shown to be ineffective against attacks constructed using expectation over the randomized components or smooth approximations of non-differentiable components [3]. Amongst the latter methods, the most successful early defense which stood the test of time was the Projected Gradient Descent (PGD) adversarial training method proposed by Madry et al. [20]. This involved minimization of cross-entropy loss on the worst-case perturbations generated using multiple iterations of constrained optimization, leading to a significantly higher computational cost when compared to standard training.

---

[*]Equal contribution.
Correspondence to: Gaurang Sriramanan <gaurangs@umd.edu>, Sravanti Addepalli <sravantia@iisc.ac.in>

35th Conference on Neural Information Processing Systems (NeurIPS 2021).

FGSM (Fast Gradient Sign Method) based adversarial training [11] alleviates the computational cost by utilizing single-step adversarial samples for training. However, this method is contingent on the local linearity assumption of the loss surface, which is often compromised during the process of FGSM training, leading to generation of weaker adversaries as training progresses. This leads to the generation of models which exhibit a false sense of robustness towards single-step attacks and are susceptible to stronger multi-step attacks. Although there have been several recent works [29, 30, 26] to circumvent gradient masking and improve the worst-case accuracy of single-step adversarial training methods, the robustness thus achieved has been suboptimal compared to multi-step defenses.

In this work, we improve the effectiveness of single-step adversarial training by introducing a novel Nuclear-Norm regularizer to impose local smoothness in the vicinity of data samples. The proposed Nuclear-Norm Adversarial Training (NuAT) utilizes batch statistics to limit the oscillation of function values in the required dimensions, thereby preventing the over-smoothing of loss surface uniformly. We summarize our contributions of our work below:

- We propose single-step Nuclear Norm Adversarial Training (NuAT), and a two step variant of the same (NuAT2), coupled with a novel cyclic-step learning rate schedule, to achieve state-of-the-art results amongst *efficient* training methods, and results comparable to the *effective* multi-step training methods.
- We demonstrate improved performance by using exponential averaging of the network weights coupled with cyclic learning rate schedule (NuAT-WA).
- We propose a two-step variant NuAT2-WA, that utilizes stable gradients from the weight-averaged model for stable initialization in the first attack step, yielding improved results.
- We improve the stability and performance of the single-step defense (NuAT) further at a marginal increase in computational cost, by introducing a hybrid approach (NuAT-H) that switches adaptively between single-step and two-step optimization for attack generation.

The organization of this paper is as follows: The preliminaries on notation and threat model are laid out in Section-2, followed by a brief note on related works in Section-3 and a discussion on challenges in single-step adversarial training in Section-4. We present a detailed description of our proposed method in Section-5, followed by experimental results to support our claims in Section-6. We conclude the paper with our analysis and future directions in Section-7.

Our code and pre-trained models are available here: `https://github.com/val-iisc/NuAT`.

## 2   Preliminaries

In this paper, we denote $x$ to be a $d$-dimensional image from an $N$-class dataset $\mathcal{D}$. Further, we denote its corresponding ground-truth label as a one-hot vector $y$. Let $f_\theta$ represent a Deep Neural Network with parameters $\theta$, that maps an input image $x$ to its pre-softmax output $f_\theta(x)$. The cross-entropy loss corresponding to the network prediction on a sample $(x, y)$ is denoted as $\ell_{CE}(f_\theta(x), y)$.

For a minibatch $B = \{(x_i, y_i)\}_{i=1}^M$, we denote $X$ as the image matrix whose $i^{th}$ row consists of flattened pixel intensities of the image $x_i$, and $Y$ as the corresponding ground-truth array. Thus, $X$ is a matrix of size $(M \times d)$, and $Y$ is a matrix of size $(M \times N)$. Let $\ell_{CE}(f_\theta(X), Y)$ denote the sum of cross-entropy losses over all data samples in the minibatch $B$. Further, for a matrix $A$, let $||A||_*$ denote the Nuclear Norm, that is the sum of the singular values of $A$.

**Adversarial Threat Model:** In this paper, we consider the robustness of Deep Networks against $\ell_\infty$ constrained adversaries. Thus under an $\varepsilon$-constraint, given a clean image $x$, an adversarially perturbed counterpart $\widetilde{x}$ can differ by at most $\varepsilon$ at any given pixel location. Further, a network $f_\theta$ is said to be $\varepsilon$-robust on a clean sample $x$ with label $y$, if $f_\theta(\widetilde{x}) = y$, for all perturbations $\widetilde{x}$ such that $||x - \widetilde{x}||_\infty \leq \varepsilon$. We note that the proposed approach can be extended to other threat models as well.

## 3   Related Works

**Multi-step Adversarial Defenses:** One of the oldest and most effective defenses known till date is the PGD adversarial training introduced by Madry et al. [20], wherein the network is trained on strong adversaries generated using multi-step optimization following a random noise initialization.

Further, Zhang et al. [33] proposed TRADES, a multi-step adversarial training method that balances the trade-off between standard accuracy and adversarial robustness. The authors proposed to train the network on adversaries generated by maximising the Kullback-Leibler (KL) Divergence, to obtain smooth models reliably and achieve state-of-the-art robust performance. Rice et al. [23] found that early stopping could be used to alleviate the phenomenon of robust-overfitting, wherein the robust performance of deep networks starts degrading after a certain number of training iterations. Strikingly, it was observed that early-stopping combined with PGD adversarial training could be used to produce significantly more robust models, which out-performed even TRADES based training. In more recent work, Pang et al. [21] discuss several implementation details with regard to the optimal choice of hyperparameters, and further improve upon the hitherto TRADES defense.

**Improving the Generalization of Defenses:** Wu et al. [31] explore the use of Adversarial Weight Perturbations (AWP) to explicitly enforce a flatter weight loss landscape. Perturbations are generated in both input space and weight space during the training regime, to maximize loss with respect to both. This is seen to significantly improve robust performance, albeit with an increased computational cost. Prior works [14] show that Stochastic Weight Averaging (SWA) can be used to train networks with flatter and wider minima to obtain improved generalisation in the context of standard training, as compared to solutions obtained using traditional Stochastic Gradient Descent (SGD). In a very recent work, Gowal et al. [12] investigate the effect of factors such as choice of activation functions, use of additional unlabelled data, as well as the application of exponential model weight averaging to boost the adversarial performance of multi-step defenses. In contrast, we propose to additionally incorporate adversaries generated from the Stochastic Weight Averaged model in order to generate stable and effective perturbations. Thus, by additionally utilising supervision from the weight-averaged model during 2-step adversarial training (NuAT2-WA), the efficacy of the defense is enhanced.

**Efficient Adversarial Training:** While multi-step adversarial methods continue to achieve the best robust performance, their computational complexity poses a severe impediment to their deployment in real-world applications that utilize large-scale datasets. Several works have been proposed to efficiently train robust models; by training without the use of adversaries during training [1], reusing gradient computations [24], or by utilising single-step adversarial training [30, 29, 26]. We expound upon the latter methods in Section-4.

**Nuclear Norm Regularization:** Optimization objectives utilizing the Nuclear Norm regularizer have seen extensive use in several computer vision, compressed sensing and machine learning applications such as in image denoising [15], low-rank matrix completion [5] and multi-task learning [22]. The Nuclear Norm provides a tight convex relaxation for the rank of a matrix, and can thus be successfully used as a surrogate objective to approximately solve intractable optimization problems. In this work, we seek to utilize Nuclear Norm regularization to enforce consistent network predictions within the vicinity of clean data samples in efficient adversarial training, as discussed in Section-5.

# 4 Analysis of Single-Step Adversarial Training

**FGSM Adversarial Training:** The efficacy of single-step approximation for generating adversarial attacks was first demonstrated by Goodfellow et al. [11]. The single-step FGSM attack was found to be sufficient to compromise the accuracy of models trained using empirical risk minimization. These single-step adversaries were further used for adversarial training (FGSM-AT), where the original dataset is augmented with some proportion of FGSM adversarial samples. However, it was later found that the resulting models were not robust to multi-step attacks due to the gradient masking effect [17]. The FGSM-AT loss formulation, which combines cross-entropy loss on clean and single-step adversarial samples, has two possible solutions. First is the desired solution, where the loss surface of the trained network continues to remain locally linear as training progresses, resulting in the generation of strong single-step adversaries, thereby facilitating the training of robust models. The second is a degenerate solution, where the model develops a convoluted loss surface, which undermines the linearity assumption required for the generation of effective single-step adversaries. Since the adversaries used for training are not effective, the adversarially trained model is not robust to multi-step attacks. This indeed results in a better minimizer of the training objective, since generating weak adversaries would result in a lower cross-entropy loss on both clean and adversarial samples.

In practice, as training progresses, the loss surface starts deviating increasingly from the first-order assumption. This reduces the efficacy of FGSM samples, leading the training process to converge

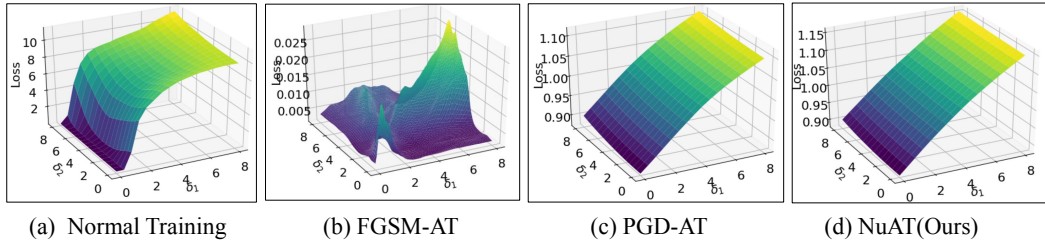

|  (a) Normal Training | (b) FGSM-AT | (c) PGD-AT | (d) NuAT(Ours) |

Figure 1: Loss surface plots showing the perturbations of the form, $x^* = x + \delta_1 g + \delta_2 g^\perp$, where $g$ is the sign of the gradient direction of the cross-entropy loss and $g^\perp$ is an orthogonal direction.

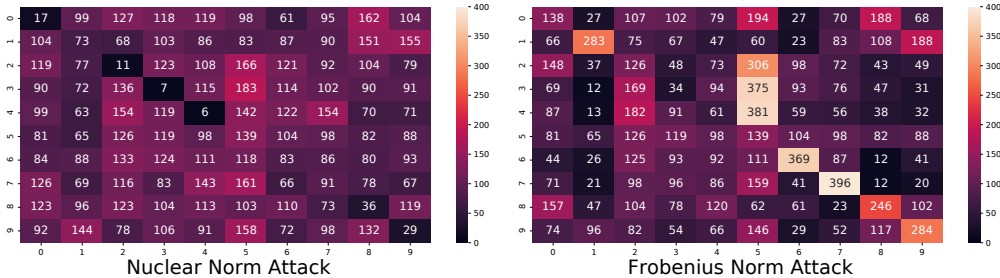

Figure 2: **Confusion Matrices** for predictions against adversarial attacks generated by maximizing the Nuclear Norm and Frobenius Norm of a matrix respectively. These are obtained for a normally trained model with ResNet-18 architecture on CIFAR-10 dataset.

to the degenerate solution. Fig.1(b) shows the loss surface of an FGSM trained model, where the gradients at the vicinity of the sample are masked, thereby preventing single-step methods from reaching a true maximum in the $\varepsilon$-ball. It can be observed that traversal along the gradient direction ($\delta_1$) does not result in increased loss, preventing the generation of strong single-step adversaries.

**R-FGSM Adversarial Training:** R-FGSM adversarial training [28] attempts to escape the gradient masking effect by computing gradients at a random point in the vicinity of the data sample. This relies on the assumption that gradients would remain constant in the local neighbourhood of data samples when the loss surface is smooth. However, this assumption can fail in case of gradient masking, leading to the generation of weak adversaries, which is termed as catastrophic overfitting in subsequent works [30]. Although Wong et al. [30] demonstrated successful R-FGSM training by adding larger magnitudes of noise before attack (FBF), and using shorter training schedules, the models trained were not competent with multi-step adversarial training methods. Also, this method does not improve robustness on larger models like WideResNet, as shown in subsequent works [26].

**Regularizers to Mitigate Gradient Masking:** One of the effective methods of mitigating gradient masking has been the use of regularizers to improve local smoothness of the loss surface [29, 26]. Sriramanan et al. [26] propose Guided Adversarial Training (GAT) to obtain significant gains in adversarial robustness when compared to R-FGSM training [30] by imposing an $\ell_2$ regularizer on the difference between softmax values of clean samples and their corresponding single-step adversaries. While this has proved to be the most effective strategy for single-step adversarial training, the results obtained are not on par with multi-step defenses. The $\ell_2$ regularizer introduced in GAT [26] attempts to uniformly limit the oscillation of the network outputs along all dimensions. While this successfully prevents the onset of gradient masking, it imposes an overly severe constraint on the outputs of the network, thereby limiting the network from learning the true underlying function. This results in a larger robustness-accuracy trade-off when compared to multi-step adversarial training. Although function smoothness is a result of PGD adversarial training as well (as shown in Fig.1(c)), the absence of an explicit regularizer allows more degrees of freedom for the function learned.

In this work, we introduce a novel Nuclear-Norm based regularizer that utilizes batch-statistics for achieving function smoothing only in the required dimensions, and to the required extent. This prevents over-smoothing of the loss function, resulting in significant performance gains. We discuss details on the proposed Nuclear-Norm Adversarial Training (NuAT) algorithm and analyse properties of models trained using this regularizer in the following section.

# 5  Proposed Method

The proposed defense incorporates Nuclear Norm based regularizer in both attack generation and adversarial training. We present details on the attack generation step in Section-5.1 followed by the adversarial training algorithm in Section-5.2. We present enhancements to the proposed defense by using two optimization steps for attack generation (Section-5.3), exponential averaging of model weights (Section-5.4) and a hybrid training approach that combines the single-step and two-step defenses (Section-5.5) in the further sections.

## 5.1  Nuclear-Norm based Attack

We consider $X$ to be the matrix composed of vectorized pixel values (arranged row-wise) of each image in a given minibatch $B$ of size $M$, $\Delta$ to be a matrix of the same dimension as $X$ consisting of independently sampled Bernoulli noise, and $Y$ to be the matrix containing the corresponding ground truth one-hot vectors. The following loss function which utilizes the pre-softmax values $\tilde{f}_\theta(.)$ is maximized for the generation of single-step adversaries:

$$\widetilde{L} = \ell_{CE}(f_\theta(X + \Delta), Y) + \lambda \cdot ||f_\theta(X + \Delta) - f_\theta(X)||_* \tag{1}$$

The attack is initialized by adding independently sampled Bernoulli noise of magnitude $\alpha$ pixel-wise to each image $(X + \Delta)$ to obtain a more reliable gradient estimate and reduce the chances of gradient masking. The first term in Eq.1 represents the cross-entropy loss on clean samples perturbed with Bernoulli noise, and the second term is the Nuclear Norm of difference between pre-softmax values of the clean images and the corresponding perturbed images. The loss in Eq.1 is maximized to find the optimal perturbation $\Delta$ within the defined $\ell_\infty$ constraint set.

While maximization of the cross-entropy term in Eq.1 aids in finding a strong attack, it is susceptible to gradient-masking effects. On the other hand, addition of the Nuclear-Norm regularizer improves the local smoothness of the loss surface, thereby preventing the onset of gradient masking and resulting in a better gradient direction for the attack. Thus, both R-FGSM (CE-only) attack and Nuclear-Norm attack play an important role during single-step adversarial training. We therefore set the coefficient of the Nuclear-Norm term to $0$ in alternate minibatches, in order to utilize R-FGSM attack and Nuclear-Norm attack alternately for training. This also diversifies the attacks used for adversarial training, as noted by Sriramanan et al. [26].

Maximization of the Nuclear Norm regularizer leads to an increase in singular values of the matrix $f_\theta(X+\Delta)-f_\theta(X)$, resulting in more diverse attacks. The generated attacks are therefore encouraged to go to a diverse set of classes, as opposed to the most confused class. We compare the confusion matrices of the predictions on attacks generated by maximizing the Nuclear Norm and Frobenius Norm respectively, with respect to the ground truth labels of the corresponding clean images in Fig.2. These are obtained for a normally trained model with ResNet-$18$ architecture on the CIFAR-10 dataset. In this experiment, we use the same class images in a single batch (with batch size $= 100$) for generating both attacks. It can be observed that the Nuclear Norm based attack results in a better diversification of the target classes when compared to the Frobenius Norm based attack. To quantify this difference, we compute the entropy of the average softmax vector for each class, and report the average entropy score across all classes. We obtain an average entropy score of $0.87$ and $0.97$ for the Frobenius Norm attack and Nuclear Norm attack respectively, indicating better diversification in the Nuclear Norm attack. It can be observed from the diagonal entries of the confusion matrices that the Nuclear Norm attack is also stronger when compared to the Frobenius Norm attack.

## 5.2  Nuclear-Norm Adversarial Training

The proposed single-step defense, Nuclear-Norm Adversarial Training (NuAT) is detailed in Algorithm-1. The following loss function is minimized during training:

$$L = \ell_{CE}(f_\theta(X), Y) + \lambda \cdot ||f_\theta(\widetilde{X}) - f_\theta(X)||_* \tag{2}$$

Here $\widetilde{X}$ is a matrix consisting of adversarial perturbations corresponding to the images in $X$. As discussed, these perturbations are generated using R-FGSM attack and Nuclear-Norm based attack in alternate iterations of training. The first term in Eq.2 corresponds to the cross-entropy loss on clean

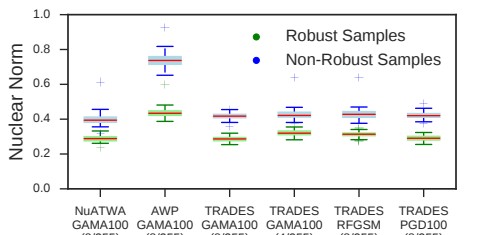 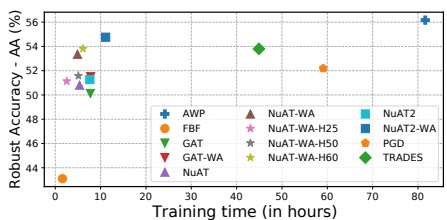

Figure 3: (a) Nuclear Norm for Robust and Non-Robust samples for multiple defense methods (b) **Computational Complexity:** Plot of Robust Accuracy on AutoAttack (AA-v2) versus training time (in hours) for multiple defense methods.

---

**Algorithm 1** Nuclear Norm Adversarial Training

---

1: **Input:** Network $f_\theta$ with parameters $\theta$, Training Data $\mathcal{D}$ with input images of dimension $d$, Minibatch Size M, Attack Size $\varepsilon$, Initial Noise Magnitude $\alpha$, Epochs $E$, Learning Rate $\eta$

2: **for** $epoch = 1$ **to** $E$ **do**

3:     **for** minibatch $B = \{(x_i, y_i)\}_{i=1}^{M} \subset \mathcal{D}$ **do**

4:        $X = \begin{bmatrix} \dots & x_1 & \dots \\ \dots & \vdots & \dots \\ \dots & x_M & \dots \end{bmatrix}, \quad \Delta = \begin{bmatrix} \dots & \delta_1 & \dots \\ \dots & \vdots & \dots \\ \dots & \delta_M & \dots \end{bmatrix}, \quad \delta_i \sim Bern^d(-\alpha, \alpha), \quad Y = \begin{bmatrix} y_1 \\ \vdots \\ y_M \end{bmatrix}$

5:        $\widetilde{L} = \ell_{CE}(f_\theta(X + \Delta), Y) + \lambda \cdot ||f_\theta(X + \Delta) - f_\theta(X)||_*$

6:        $\Delta = \Delta + \varepsilon \cdot \text{sign}\left(\nabla_\Delta \widetilde{L}\right)$

7:        $\Delta = Clamp(\Delta, -\varepsilon, \varepsilon), \quad \widetilde{X} = Clamp(X + \Delta, 0, 1)$

8:        $L = \ell_{CE}(f_\theta(X), Y) + \lambda \cdot ||f_\theta(\widetilde{X}) - f_\theta(X)||_*$

9:        $\theta = \theta - \frac{1}{M} \cdot \eta \cdot \nabla_\theta L$

10:     **end for**

11: **end for**

---

samples, and the second term corresponds to Nuclear-Norm of difference in pre-softmax values of clean images $X$ and their corresponding single-step adversaries $\widetilde{X}$. We utilize the same loss function for both adversary generation and training as used by Madry et al. [20]. The proposed regularizer in Eq.1 and Eq.2 is weighted by a factor $\lambda$, which controls the accuracy-robustness trade-off. We use the same weight $\lambda$ for both adversary generation and training, which is linearly increased over the training epochs. We show the sensitivity of training to the parameter $\lambda$ in the Supplementary material. Further, we use a novel cyclic-step learning rate schedule that incorporates the cyclic schedule in early epochs of training, and transitions to the use of a step schedule towards the end (Details in the Supplementary Section).

While the Nuclear Norm attack encourages the generation of perturbations which maximize the singular values of the matrix $f_\theta(\widetilde{X}) - f_\theta(X)$, the Nuclear Norm Adversarial Training (NuAT) attempts to minimize the same. The Nuclear Norm of a matrix forms a uniform upper bound of its Frobenius norm (Theorem-1 in the Supplementary), and therefore the proposed defense minimizes the difference between the predictions of the clean and adversarial images, encouraging smoothing of the loss surface. Further, it can be shown that the Nuclear Norm is the lower convex envelope of the rank of a matrix (Theorem-2 in the Supplementary). Since any function and its lower convex envelope have the same global minimizer (Lemma-2 in the Supplementary), Nuclear Norm minimization helps induce minimality in the rank of the matrix, thereby encouraging associations between the columns (or classes) of the matrix (data). This allows the samples within a batch to aid each other and utilize the relationship between the oscillation of the function values across different dimensions (classes) to obtain a better robustness-accuracy trade-off, as shown in Section-6.

As shown in Fig.3(a), across multiple robust models, lower Nuclear Norm values correspond to robust samples and higher values correspond to non-robust ones. This reiterates our hypothesis that explicit minimization of the Nuclear Norm regularizer can improve the adversarial robustness of models.

### 5.3 Two-step Attack Optimization

While the proposed single-step defense NuAT mitigates gradient masking effectively and achieves an improvement over existing efficient defenses (Section-6), we propose to enhance its performance further by using two steps of optimization during attack generation in the NuAT2 defense. In the first step, we use the same loss as the single-step attack (Eq.1), whereas in the second step of optimization, we use only cross-entropy loss, which corresponds to the setting $\lambda = 0$. Similar to the GAMA attack [26], the first step involving the use of Nuclear-Norm relaxation term generates a more reliable gradient direction by smoothing the loss surface, and the cross-entropy objective of the second step is optimal for fooling the classifier. We use a step-size of $\varepsilon$ for each of the steps, in order to allow the best of both objectives to drive the generation of the attack.

### 5.4 Model Weight Averaging

Stochastic Weight Averaging [14, 7] is known to produce flatter and wider optima when compared to Stochastic Gradient Descent (SGD) leading to better generalization. We utilize exponential averaging of model weights [7, 12] along with a cyclic learning rate schedule to boost the performance of the proposed single-step defense further without an increase in computational cost. We refer to the single-step defense incorporating model weight averaging as NuAT-WA. Additionally, in the two-step defense NuAT2-WA, we utilize gradients from the weight averaged model in the first attack step. This results in a more reliable gradient direction initially since the weight averaged model is more stable with a smoother loss surface when compared to the main model which is being trained. As shown in Fig.3(b), the NuAT2-WA defense outperforms even multi-step adversarial defenses such as PGD-AT [20] and TRADES [33] at a significantly lower computational cost. We present further details on the NuAT2-WA defense in the Supplementary.

### 5.5 Hybrid Nuclear Norm Adversarial Training

In this work we propose two main variants of the NuAT defense, single-step and two-step. While single-step training results in the most effective defenses at low compute, two-step training is more stable, and yields results which are close to even the state-of-the-art multi-step (10-11 step) training methods. As observed in prior works [30, 19], single-step defenses cannot be used for long training regimes due to issues relating to stability of training. While the proposed NuAT single-step defense is significantly more stable compared to such works [30], it is not as stable as the two-step defense for large model capacities and longer training schedules. This limitation in the number of training epochs curtails the optimal performance that can be achieved by the base algorithm.

Li et al. [19] use a validation set to detect the onset of instability in training and switch to a 10-step defense for a few iterations. Inspired by this, we propose to adaptively switch between the proposed single-step (NuAT) and two-step (NuAT2) defenses by tracking a sudden drop in the training loss on clean samples when compared to the average loss of the previous epoch. While Li et al. [19] required a separate validation set and 10-step training, we seamlessly switch between the proposed single-step and two-step defenses without any additional overheads. By varying the threshold at which we switch from single-step to two-step attacks, this method can either be used for improving the stability of the single-step run, or to additionally boost the performance by switching to stronger two-step attacks more often. For the hybrid NuAT-H and NuAT-WA-H models so trained, the additional epochs incurred are presented as a subscript in Table-1b. We thereby obtain significantly more robust models, while requiring an additional overhead of only upto 6 training epochs. We further expound upon adaptive switching integrated with NuAT in the Supplementary Section.

## 6 Experiments and Analysis

**Overview of Datasets and Evaluations**: We run extensive evaluation on the following three benchmark datasets: MNIST [18], CIFAR-10 [16] and a 100 class subset of ImageNet [10, 26]. For each dataset, we maintain a train-validation split that is balanced equally across all classes. We present details on the datasets, train-validation splits and model architectures in the Supplementary Section. In this Section, we present evaluations on a wide range of adversaries in a white-box setting. We consider an $\ell_\infty$ adversarial constraint of $\varepsilon = 8/255$ for CIFAR-10 and ImageNet-100, and of $\varepsilon = 0.3$ for MNIST, as is standard. We first present results using untargeted attacks, over a broad range of

Table 1: **CIFAR**-10 **White-box evaluation:** Accuracy (%) of various defenses (rows) against different attacks for the CIFAR-10 dataset, under an $\ell_\infty$ constraint of $\varepsilon = 8/255$. The defenses presented are as follows: FGSM-AT [11], RFGSM-AT [28], ATF [24], FBF [30], R-MGM [29], GAT [26], PGD-AT [20, 23], TRADES [33, 21] and AWP [31].

(a) **ResNet-18 architecture**

| Method | # AT steps | Clean Acc | PGD (n-steps) 20 | 500 | GAMA 100 | AA (v1) |
|---|---|---|---|---|---|---|
| Normal | 0 | 92.30 | 0.00 | 0.00 | 0.00 | 0.00 |
| FGSM-AT | 1 | **92.89** | 0.00 | 0.00 | 0.00 | 0.00 |
| RFGSM-AT | 1 | 89.24 | 35.02 | 34.17 | 33.87 | 33.16 |
| ATF | 1 | 71.77 | 43.53 | 43.52 | 40.34 | 40.22 |
| FBF | 1 | 82.83 | 46.41 | 46.03 | 43.85 | 43.12 |
| R-MGM | 1 | 82.29 | 46.23 | 45.79 | 44.06 | 43.72 |
| GAT | 1 | 80.49 | 53.13 | 53.08 | 47.76 | 47.30 |
| GAT-WA | 1 | 79.47 | **54.40** | **54.37** | 49.00 | 48.28 |
| NuAT (**Ours**) | 1 | 81.01 | 53.30 | 52.97 | 49.46 | 49.24 |
| NuAT-WA (**Ours**) | 1 | 82.21 | 54.14 | 53.95 | **50.97** | **50.75** |
| PGD-AT | 10 | 81.12 | 53.08 | 52.89 | 49.08 | 48.75 |
| TRADES | 10 | 81.47 | 52.73 | 52.61 | 49.22 | 49.06 |
| TRADES-WA | 10 | 80.19 | 52.98 | 52.88 | 49.49 | 49.39 |
| AWP | 11 | **81.99** | **55.60** | **55.52** | **51.65** | **51.45** |

(b) **WideResNet-34-10 architecture**

| Method | AT-steps (epochs) | Clean Acc | PGD 100 | GAMA 100 | AA (v2) |
|---|---|---|---|---|---|
| FBF | 1 (30) | 82.05 | 45.57 | 43.13 | 43.10 |
| GAT | 1 (85) | 85.17 | 55.12 | 50.76 | 50.12 |
| GAT-WA | 1 (85) | 84.61 | 57.28 | 52.19 | 51.50 |
| Variants of NuAT (**Ours**) | | | | | |
| NuAT | 1 (55) | 85.30 | 53.82 | 51.34 | 50.81 |
| NuAT-H | 1 ($50_{+5}$) | 84.58 | 54.89 | 51.93 | 51.58 |
| NuAT-WA | 1 (50) | 85.29 | 56.21 | 53.73 | 53.36 |
| NuAT-WA-H | 1 ($25_{+2}$) | 81.98 | 54.82 | 51.41 | 51.14 |
| NuAT-WA-H | 1 ($60_{+6}$) | 84.93 | 57.51 | 54.28 | 53.81 |
| NuAT2 | 2 (55) | 84.76 | 54.50 | 51.99 | 51.27 |
| NuAT2-WA | 2 (80) | **86.32** | 57.74 | 55.08 | 54.76 |
| NuAT-WA-H | 1 ($80_{+32}$) | 85.40 | **58.25** | **55.35** | **54.96** |
| TRADES | 10 (110) | 85.48 | 56.35 | 53.88 | 53.80 |
| PGD | 10 (200) | **86.07** | 55.74 | 52.70 | 52.19 |
| AWP | 11 (200) | 85.36 | **59.13** | **56.35** | **56.17** |

adversaries such as PGD [20] and GAMA-PGD [26]. We additionally present results on AutoAttack [9], consisting of an ensemble of the APGD-CE, APGD-DLR, FAB [8] and Square [2] attacks. We remark that the GAMA-PGD and AA attacks comprise the strongest known attacks till date, and provide a reliable estimate of the true robustness of Deep Networks.

## 6.1 White-Box Evaluations

**CIFAR-10 (ResNet-18):** In Table-1a, we present an evaluation of the proposed method on the CIFAR-10 dataset, and compare it with several existing defenses using the ResNet-18 architecture. We present single-step defenses in the first partition of the table, and multi-step defenses in the second. We observe that the proposed method NuAT performs significantly better than existing single-step training methods such as FGSM-AT [11], RFGSM-AT [28], ATF [24], FBF [30], R-MGM [29] and GAT [26]. Further, we note that despite using only single-step adversaries for training, NuAT performs better than multi-step defenses such as PGD adversarial training (PGD-AT) [20, 23] and TRADES [33, 21]. We observe that by incorporating exponential weight-averaging [14] with the proposed single-step defense, NuAT-WA is seen to achieve enhanced robust performance, along with improved clean accuracy. Further on both these metrics, NuAT-WA significantly outperforms other methods with weight-averaging incorporated such as GAT-WA and TRADES-WA. We also note that NuAT-WA achieves robust accuracy comparable to that of AWP [31], while requiring only a fraction of the budget to generate adversaries during training. Over five runs with random seeds, we record a standard deviation of 0.11 and 0.16 respectively for the clean and 100-step GAMA-PGD accuracies.

**CIFAR-10 (WRN-34-10):** In Table-1b, we compare the proposed defense with existing approaches on the WideResNet-34-10 architecture. We note that that NuAT significantly outperforms the FBF model, since the latter suffers from catastrophic overfitting on larger model capacities. We also remark that NuAT achieves better robustness, and is able to strike a slightly more favourable robustness-accuracy trade-off as compared to GAT, while requiring far fewer epochs for training as well. Next, we introduce 2-step Nuclear Norm Adversarial Training (NuAT2), wherein the first step is generated in a manner identical to that of NuAT, and followed by the maximisation of cross-entropy loss in the second. This procedure generates diverse adversaries and is seen to significantly improve robustness. We further present evaluations on Hybrid training with NuAT, wherein the number of effective epochs which utilize two-step attack generation (rounded to the next largest integer) are indicated as a subscript in Table-1b. We thus observe that NuAT-H effectively bridges the gap between NuAT and NuAT-2, while requiring an additional effective overhead of only 5 training epochs.

**Weight Averaging in Wide Networks:** Indeed, in Table-1b, we observe a remarkable improvement with NuAT-WA, while requiring little computational overhead. The NuAT-WA model is seen to significantly outperform 10-step PGD adversarial training, and achieve robustness comparable to that of TRADES while requiring significantly less computational time for training. We also present results by incorporating supervision from the exponentially weight-averaged model with 2-step

Table 2: **Consolidated results against white-box:** Prediction accuracy (%) of different models against PGD 500-step, GAMA-PGD 100-step, AutoAttack-v1 (AA) attacks across CIFAR-10, ImageNet-100 and MNIST datasets. The defenses presented are: RFGSM-AT [28], FBF [30], R-MGM [29], GAT [26], PGD-AT [20, 23], TRADES [33, 21] and AWP [31].

| | CIFAR-10 | | | | ImageNet-100 | | | | MNIST | | | |
|---|---|---|---|---|---|---|---|---|---|---|---|---|
| | Clean Acc | PGD 500 | GAMA 100 | AA (v1) | Clean Acc | PGD 500 | GAMA 100 | AA (v1) | Clean Acc | PGD 500 | GAMA 100 | AA (v1) |
| Normal | **92.30** | 0.00 | 0.00 | 0.00 | **81.44** | 0.00 | 0.00 | 0.00 | 99.20 | 0.00 | 0.00 | 0.00 |
| RFGSM-AT | 89.24 | 34.17 | 33.87 | 33.16 | 78.46 | 13.88 | 13.38 | 12.96 | **99.37** | 85.32 | 83.64 | 82.28 |
| FBF | 82.83 | 46.03 | 43.85 | 42.37 | 57.32 | 27.22 | 21.78 | 20.66 | 99.30 | 91.37 | 87.27 | 79.02 |
| R-MGM | 82.29 | 45.79 | 44.06 | 43.72 | 64.84 | 31.68 | 27.46 | 27.68 | 99.04 | 90.56 | 88.13 | 86.21 |
| GAT | 80.49 | 53.08 | 47.76 | 47.30 | 67.98 | 37.46 | 29.30 | 28.92 | **99.37** | 94.44 | 92.96 | 90.62 |
| GAT-WA | 79.47 | **54.37** | 49.00 | 48.28 | 68.02 | 38.02 | 30.16 | 29.30 | 99.36 | 94.76 | 93.25 | 91.48 |
| NuAT (**Ours**) | 81.01 | 52.97 | 49.46 | 49.24 | 69.00 | 37.60 | 32.38 | 31.96 | **99.37** | 96.24 | 94.65 | **93.11** |
| NuAT-WA (**Ours**) | 82.21 | 53.95 | **50.97** | **50.75** | 68.40 | **38.68** | **33.22** | **33.16** | 99.36 | **96.30** | **94.70** | 93.10 |
| TRADES | 81.47 | 52.61 | 49.22 | 49.06 | 62.88 | 37.24 | 31.44 | 31.66 | 99.32 | 93.40 | 92.74 | 92.19 |
| PGD-AT | 81.12 | 52.89 | 49.08 | 48.75 | 68.62 | 36.56 | 32.24 | 32.98 | 99.27 | 93.98 | 92.80 | 91.81 |
| AWP | 81.99 | **55.52** | **51.65** | **51.45** | 64.84 | 36.04 | 29.58 | 29.16 | 99.12 | 94.90 | 94.19 | **93.47** |

Nuclear Norm Adversarial Training (NuAT2-WA) to enhance attack strength and diversity. Here, we first generate a single-step attack by maximising the proposed loss (Eq.1) on the exponential weight-averaged model, and then utilise a second step that maximizes cross-entropy loss on the original base-model to obtain strong adversaries. NuAT2-WA amply surpasses the TRADES defense both in terms of robust performance, as well as clean accuracy, and achieves robustness comparable to that of 11-step AWP training. We discuss the training methodology in greater detail and present similar detailed results on ImageNet-100 and MNIST in the Supplementary Material. We also similarly observe significant improvement in robust performance by incorporating exponential weight-averaging in Hyrbrid NuAT training. We also note that Hybrid training enables longer training schedules (such as 112 effective epochs) while maintaining stability akin to that of multi-step defenses, thereby achieving striking gains in robust accuracy. We expound upon this further and present additional evaluations in the Supplementary Section.

**Consolidated results across three datasets:** In Table-2, we present consolidated results across all three datasets, CIFAR-10, ImageNet-100 and MNIST using the PGD 500-step attack, GAMA-PGD attack and AutoAttack, which comprises of the strongest known white-box attacks to date. We primarily utilise the baseline models as introduced by Sriramanan et al. [26] to perform a comparative analysis for the latter two datasets. We observe that the proposed approach achieves significantly improved robust accuracy against GAMA-PGD attack and AutoAttack [9] on the MNIST dataset, compared to even multi-step defenses such as TRADES and PGD-AT. We further remark that NuAT achieves this without a drop in standard accuracy on clean samples. We also observe again that NuAT achieves enhanced GAMA-PGD and AutoAttack accuracy as compared to all other defense methods (including AWP [31]) on the highly challenging ImageNet-100 dataset, while requiring significantly less training time, demonstrating the scalability of the proposed NuAT defense. Lastly, we remark that exponential weight-averaging provides a significant boost in robust performance on ImageNet-100, on all three white-box attacks. In summary, NuAT outperforms existing single-step defenses, with improvements in AutoAttack accuracy of 1.94%, 3.04% and 2.49% over the previous state-of-the-art defense GAT [26] on the CIFAR-10, ImageNet-100 and MNIST datasets respectively. Similarly, NuAT-WA significantly improves upon GAT-WA, by 2.47%, 3.86% and 1.62% respectively on the CIFAR-10, ImageNet-100 and MNIST datasets. In addition, the proposed defense achieves enhanced or comparable performance to that of multi-step methods, for a fraction of the computational cost.

**Scalability to ImageNet:** While the scalability of single-step defenses to large capacity models such as WideResNet-34-10 has been a bigger challenge than scaling to larger datasets such as ImageNet (as observed in prior works [30]), training robust models is nonetheless exigent in the latter setting. Here we analyze the efficacy of the proposed defense on the original ImageNet dataset [10], composed of 1000 classes with over 1.2 million images. Using the ResNet-50 architecture under the $\ell_\infty$ constraint set with $\varepsilon = 4/255$, NuAT-WA achieves 59.65% clean accuracy and 35.42% against GAMA-PGD, significantly improving upon FBF [30] which achieves 56.70% and 27.44% respectively on clean samples and the GAMA-PGD attack. We expound further upon this in the Supplementary Section.

## 6.2 Computational Complexity

We compare the computational complexity of the proposed defense with various other efficient and effective defenses on the WideResNet-34-10 architecture for CIFAR-10 dataset in Fig.3(b). We report accuracy against the latest version (v2) of AutoAttack, which constitutes one of the strongest known ensemble of attacks till date. While the fastest defense amongst all is FBF [30], it achieves a significantly lower robust accuracy when compared to the other defenses. On the other extreme is AWP [31], which has the highest robust accuracy, but comes with a very high computational cost ($8\times$ that of NuAT2-WA) . The proposed defense NuAT2-WA achieves a robust accuracy of $54.76\%$, outperforming all other defenses (even TRADES, PGD) at a significantly lower computational cost.

## 6.3 Loss Surface Plots

We visualize the loss surface plots for models trained using different training methods in Fig.1. We plot the cross-entropy loss by varying the perturbation along two axes: one along the sign of the gradient, and another along a random direction orthogonal to the gradient. As expected, we observe that FGSM adversarial training produces extreme gradient masking in Fig.1(b), resulting in a convoluted loss landscape. On the other hand, the loss surface of the proposed single-step defense Nuclear Norm Adversarial Training (NuAT) is seen to be smooth in Fig.1(d), indicating the absence of gradient masking. We further note that the smoothness of loss surface of NuAT is comparable to that of PGD-AT, which is a multi-step adversarial training method.

## 6.4 Sanity Checks to Verify the Absence of Gradient Masking

We further present detailed evaluations on several other targeted and untargeted white-box attacks, black-box attacks and gradient-free attacks in the Supplementary Section to verify the robustness of the single-step NuAT defense over all three datasets. Furthermore, we also present adaptive attacks that are cognizant of the proposed defense and other sanity checks in the Supplementary Material to verify the absence of gradient masking in the proposed defense as laid out by Athalye et al. [3].

# 7 Conclusions

In this work, we propose a novel Nuclear Norm regularizer to improve the adversarial robustness of Deep Networks through the use of single-step adversarial training. Training with the proposed Nuclear Norm regularizer enforces function smoothing in the vicinity of clean samples by incorporating joint batch-statistics of adversarial samples, thereby resulting in enhanced robustness. We demonstrate that the proposed NuAT defense out-performs existing single-step adversarial training methods, and scales well to large model capacities and large-scale datasets such as ImageNet-100. Additionally, we integrate exponential weight-averaging to achieve significantly improved robust performance. We further introduce a two-step variant of the proposed defense that helps incorporate supervision from the weight-averaged model during training, resulting in models that achieve near state-of-the-art adversarial robustness at a significantly lower computational cost. Finally, we present Hyrbrid NuAT that effectively bridges the computation-accuracy trade-off by adaptively switching between single-step and two-step training, to obtain significantly more robust and stable models while requiring only a marginal increase in computational cost.

# 8 Limitations and Societal Impact

While the proposed Nuclear Norm attack is indeed novel and can perhaps be used to compromise Deep Networks, it is limited to a white-box setting and is appreciably weaker than existing iterative adversarial attacks such as AutoAttack [9] or GAMA-PGD [26]. The proposed defense method on the other hand facilitates the effective training of robust networks through efficient means, and thus has immense potential to have a positive impact on society. We are also hopeful that future work could extend the proposed approach to other adversarial threat models of interest, improve upon the robustness-accuracy trade-off incurred, and present provable robustness certificates to further justify worst-case performance. Furthermore, the proposed approach could be significantly strengthened if the absence of gradient masking could be substantiated from a theoretical standpoint, with convergence guarantees for single-step adversarial training in practical settings.

# 9 Acknowledgments and Disclosure of Funding

This work was supported by Uchhatar Avishkar Yojana (UAY) project (IISC 10), MHRD, Govt. of India. Sravanti Addepalli is supported by a Google PhD Fellowship in Machine Learning. We are thankful for the support.

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
