# Supplementary Material: Towards Efficient and Effective Adversarial Training

**Gaurang Sriramanan**\*, **Sravanti Addepalli**\*, **Arya Baburaj**, **R.Venkatesh Babu**
Video Analytics Lab, Department of Computational and Data Sciences
Indian Institute of Science, Bangalore, India

## 1   Analysing the Nuclear Norm Objective

We first prove that the Nuclear norm forms a uniform upper-bound of the Frobenius norm of a matrix:

**Theorem 1:** Suppose $A$ is any real/complex $m \times n$ matrix. Then $||A||_* \geq ||A||_F$.

**Proof:** Let $A = U\Lambda V^T$, be the Singular Value Decomposition of $A$, where $U, V$ are Unitary square matrices of size $m \times m$ and $n \times n$ respectively, and $\Lambda$ is a $m \times n$ rectangular diagonal matrix with non-negative entries $\sigma_1, \sigma_2, \ldots$, which are the singular values of $A$.
As is standard convention, we assume descending order of singular values. Further, if $\rho$ is the rank of matrix $A$, then $\sigma_i = 0$ for all $i > \rho$. Then, by definition, $||A||_* := \sum_{i=1}^{\min m,n} \sigma_i = \sum_{i=1}^{\rho} \sigma_i$.

We now observe that

$$||A||_*^2 = \left(\sum_{i=1}^{\rho} \sigma_i\right)^2 = \sum_{i=1}^{\rho} \sigma_i^2 + \sum_{i \neq j} \sigma_i \cdot \sigma_j \geq \sum_{i=1}^{\rho} \sigma_i^2$$

where the inequality follows from the non-negativity of singular values. We further observe that

$$\sum_{i=1}^{\rho} \sigma_i^2 = ||\Lambda||_F^2 = ||U\Lambda||_F^2 = ||U\Lambda V^T||_F^2 = ||A||_F^2$$

since $U, V$ are Unitary matrices which are $\ell_2$-norm isometries by definition, and thus do not affect the $\ell_2$-norms of the vectors comprising the rows and columns of $\Lambda$. This can also be seen intuitively, as Unitary matrices correspond precisely to the family of reflection and rotation matrices, and it can be seen fairly clearly that the Frobenius norm is invariant to such transformations since the scaling is unaffected in the process. ∎

Let $\overrightarrow{\sigma_A}$ represent the vector of the singular values of $A$. We then observe that the Nuclear norm of matrix $A$ is the $\ell_1$ norm of $\overrightarrow{\sigma_A}$. That is, $||A||_* = \sum_{i=1}^{\rho} \sigma_i = ||\overrightarrow{\sigma_A}||_1$ .

Thus, akin to the manner in which sparsity is induced in a vector through $\ell_1$ norm based regularization, the minimisation of Nuclear Norm induces sparsity in terms of the number of non-zero singular values, which is precisely the rank $\rho$ of the matrix.

**Definition:** Given a real-valued function $f : \mathcal{S} \longrightarrow \mathbb{R}$, where $\mathcal{S}$ is a convex set, the lower convex envelope $f^\cup$ is defined as:

$$f^\cup = \sup_{g}\{ g \mid g \text{ is convex over } \mathcal{S}, \ g(x) \leq f(x) \ \forall x \in \mathcal{S}\}.$$

**Lemma 1:** Suppose $f_1, f_2 : \mathcal{S} \longrightarrow \mathbb{R}$ are convex functions over a convex set $\mathcal{S}$. Then $f(x) := \max\{f_1(x), f_2(x)\}$ is also a convex function.

---

\*Equal contribution.
Correspondence to: Gaurang Sriramanan <gaurangs@iisc.ac.in>, Sravanti Addepalli <sravantia@iisc.ac.in>

35th Conference on Neural Information Processing Systems (NeurIPS 2021).

**Proof:** Let $x, y \in \mathcal{S}$. Then:

$$
\begin{aligned}
f(\lambda x + (1-\lambda)y) &= f_i(\lambda x + (1-\lambda)y) \quad \text{(for } i = 1 \text{ or } 2) \\
&\leq \lambda f_i(x) + (1-\lambda)f_i(y) \\
&\leq \lambda \max\{f_1(x), f_2(x)\} + (1-\lambda)\max\{f_1(y), f_2(y)\} \\
&= \lambda f(x) + (1-\lambda)f(y)
\end{aligned}
$$

$\blacksquare$

A pertinent property of the lower convex envelope of a function is that they have the same minimizer:

**Lemma 2:** Suppose $x^* \in \underset{x}{\operatorname{argmin}}\{\ f(x) \mid x \in \mathcal{S}\}$. Then $\underset{x}{\min}\{\ f^\cup(x) \mid x \in \mathcal{S}\} = f^\cup(x^*)$ and $f^\cup(x^*) = f(x^*)$.

**Proof:** Since $f^\cup(x) \leq f(x)$ for all $x \in \mathcal{S}$, $\underset{x \in \mathcal{S}}{\min} f^\cup(x) \leq \underset{x \in \mathcal{S}}{\min} f(x)$.

We shall prove the converse inequality using contradiction: assume that $f^\cup(x^*) < f(x^*)$.
Define a constant function $g(x) = f(x^*)$ for all $x \in \mathcal{S}$, which is trivially convex over $\mathcal{S}$. Now, define $\widetilde{f}(x) = \max\{g(x), f^\cup(x)\}$, which is a convex function by Lemma 1. Further, from the definition of $\widetilde{f}$, we see that $\widetilde{f}(x) \leq f(x)$ for all $x \in \mathcal{S}$, but $f^\cup(x^*) < f(x^*) = \widetilde{f}(x^*)$. This contradicts the fact that $f^\cup$ is a lower convex envelope, since $f^\cup = \sup\{\ g \mid g \text{ is convex over } \mathcal{S}, \ g(x) \leq f(x) \ \forall x \in \mathcal{S}\}$, and $\widetilde{f}$ is one such candidate function in the supremum. Thus, $f^\cup(x^*) = f(x^*)$. $\blacksquare$

Furthermore, Fazel [5] shows that the Nuclear Norm is the tightest convex relaxation of the rank of a matrix, for a given constraint on the Spectral Norm of the matrix:

**Definition:** The Spectral Norm of a matrix $A$ is defined as $||A||_\infty = \underset{i}{\max} \sigma_i$.

**Theorem 2 (Fazel [5]):** Suppose $\mathcal{S}_r = \{A \mid ||A||_\infty \leq r\}$, and let $\rho(A) : \mathcal{S}_r \longrightarrow \mathbb{Z}$ denote the rank of matrix $A$. Then the lower convex envelope of $\rho(A)$ over $\mathcal{S}_r$ is $\rho^\cup(A) = \frac{1}{r}||A||_*$.

We thus find that a minimizer of the rank of the matrix is one which additionally minimizes the Nuclear Norm, since any function and its lower convex envelope have the same global minimizer.

## 2 Details on Training Algorithms

### 2.1 Details on NuAT2-WA training algorithm

The proposed two-step training algorithm coupled with weight averaging (NuAT2-WA) is presented in Algorithm-1. Similar to NuAT2 training as presented above, in the first attack step we use a combination of cross-entropy loss and the Nuclear-Norm regularizer, while in the second step only the cross-entropy loss is maximized. However the crucial difference of NuAT2-WA with respect to NuAT2 is the use of the weight averaged model for generation of the first attack step as shown in L-7 of Algorithm-1. This results in a more reliable gradient direction initially since the weight averaged model is more stable with a smoother loss surface when compared to the main model which is being trained. In order to highlight the significance of using the weight averaged model in the first step of training, we perform an ablation experiment by using the main model for both attack steps (**A1**). As shown in Table-1, the proposed NuAT2-WA algorithm results in a significantly better clean and robust accuracy for the same training budget. Further, we note that using the proposed algorithm we achieve comparable results even for a significantly lower number of training epochs (**A2**).

The weights of the NuAT2-WA model are computed after every iteration using exponential weight averaging as shown in L13 of Algorithm-1. It is to be noted that weight averaging does not increase the computational complexity of training (Fig.3(b) of the Main Paper). The optimal value of $\tau$ in most cases is $0.9998$ for a training batch size of $64$. We select the best model based on I-FGSM 7-step accuracy on a hold out validation set as detailed in Sec-3.

### 2.2 Details on Hybrid NuAT training algorithm

In this work we propose two main variants of the NuAT defense, single-step and two-step. While single-step training results in the most effective defenses at low compute, two-step training is more

**Algorithm 1** NuAT2-WA: 2-Step Weight Averaged Nuclear Norm Adversarial Training

1: **Input:** Network $f_\theta$ with parameters $\theta$, Training Data $\mathcal{D}$ with input images of dimension $d$, Minibatch Size M, Attack Size $\varepsilon$, Initial Noise Magnitude $\alpha$, Learning Rate $\eta$, Epochs $E$, Exponential Weight Averaging factor $\tau$

2: **Initialize** $\omega = \theta$

3: **for** $epoch = 1$ **to** $E$ **do**

4:     **for** minibatch $B = \{(x_i, y_i)\}_{i=1}^M \subset \mathcal{D}$ **do**

5:       $X = \begin{bmatrix} \dots & x_1 & \dots \\ \dots & \vdots & \dots \\ \dots & x_M & \dots \end{bmatrix}, \quad \Delta = \begin{bmatrix} \dots & \delta_1 & \dots \\ \dots & \vdots & \dots \\ \dots & \delta_M & \dots \end{bmatrix}, \quad \delta_i \sim Bern^d(-\alpha, \alpha), \quad Y = \begin{bmatrix} y_1 \\ \vdots \\ y_M \end{bmatrix}$

6:       $\widetilde{L} = \ell_{CE}(f_\omega(X + \Delta), Y) + \lambda \cdot ||f_\omega(X + \Delta) - f_\omega(X)||_*$

7:       $\Delta = \Delta + \varepsilon \cdot \text{sign}\left(\nabla_\Delta \widetilde{L}\right)$

8:       $\Delta = Clamp\,(\Delta, -\varepsilon, \varepsilon)$

9:       $\Delta = \Delta + \varepsilon \cdot \text{sign}(\nabla_\Delta \ell_{CE}(f_\theta(X + \Delta), Y))$

10:      $\Delta = Clamp\,(\Delta, -\varepsilon, \varepsilon), \quad \widetilde{X} = Clamp\,(X + \Delta, 0, 1)$

11:      $L = \ell_{CE}(f_\theta(X), Y) + \lambda \cdot ||f_\theta(\widetilde{X}) - f_\theta(X)||_*$

12:      $\theta = \theta - \frac{1}{M} \cdot \eta \cdot \nabla_\theta L$

13:      $\omega = (1 - \tau) * \theta + \tau * \omega$

14:     **end for**

15: **end for**

Table 1: **Ablation experiments** to highlight the importance of using Weight-Averaged model for **attack generation in NuAT2-WA**. Prediction accuracy (%) on the original dataset (Clean) and robust accuracy (%) against two white-box attacks are presented using WideResNet-34-10 architecture on CIFAR-10 dataset.

| Training Method | Epochs | Clean | FGSM | GAMA-100 |
|---|---|---|---|---|
| NuAT2-WA (**Ours**) | 80 | **86.32** | **63.48** | **55.08** |
| **A1:** Without using WA model for attack | 80 | 83.48 | 63.18 | 54.50 |
| **A2:** NuAT2-WA (Shorter schedule) | 55 | 85.62 | 62.97 | 54.61 |

stable, and yields results which are close to even the state-of-the-art multi-step (10-11 step) training methods. In order to combine the benefits of both approaches, we propose the Hybrid training algorithm that adaptively switches between one-step and two-step attacks for training.

We start with the default setting of using single-step attacks for training. As training progresses, the model may start generating weaker single-step attacks, which results in a sudden drop in both cross-entropy loss on clean samples, and the Nuclear Norm loss. While the Nuclear Norm regularizer overcomes such instability issues in single-step training to a large extent, this issue limits the training of large capacity models to short training schedules. We mitigate this issue by detecting it early, and dynamically switching to stronger attacks as proposed by Li et al. [10]. Different from their approach which uses validation set accuracy for detection, we propose to detect this without additional computational overheads. This is done by comparing the loss on clean samples in the current iteration with the average loss on clean samples in the previous epoch. If the ratio of the loss in current iteration to the average loss in the previous epoch is below a preset threshold $t$, the algorithm switches from a one-step attack to a two-step attack for all images in the current iteration. We keep track of the number of iterations where the algorithm switches to a two-step attack in order to report the overhead in computation time when compared to a single step defense.

For the cyclic learning rate schedule, we apply this detection during the ramp-down period when the learning rate decreases from a peak value to 0. We start with a threshold $t_{min}$ and ramp up linearly to a threshold $t_{max}$ since the stability issues are more likely to occur in late stages of training, at lower learning rates. We note that such stabilization is not required for lower values of $\lambda$ and for shorter training schedules. However, higher $\lambda$ values and longer training schedules can boost the performance of the proposed method further as shown in Table-2.

Table 2: **Hybrid NuAT:** Accuracy (%) of Hyrbid NuAT models trained with different thresholds and epoch-budgets on the CIFAR-10 dataset. Prediction accuracy (%) on the original dataset (Clean) and robust accuracy (%) against GAMA PGD 100-step white-box attack are presented.

| Epochs | Net Epochs | Threshold $t_{min}, t_{max}$ | Clean (Base-model) | GAMA-100 (Base-model) | Clean (WA model) | GAMA-100 (WA model) |
|---|---|---|---|---|---|---|
| $50_{\ +5}$ | 55 | 0.7 , 1.0 | 84.58 | 51.93 | 84.41 | 53.69 |
| $60_{\ +6}$ | 66 | 0.7 , 1.0 | 86.29 | 49.86 | 84.93 | 54.28 |
| $70_{\ +18}$ | 88 | 0.9 , 1.0 | 86.65 | **52.05** | 85.31 | 54.79 |
| $80_{\ +32}$ | 112 | 0.9 , 1.1 | **87.19** | 50.85 | 85.40 | **55.35** |
| $90_{\ +47}$ | 137 | 0.9 , 1.2 | 86.18 | 51.47 | 85.43 | 55.18 |
| $100_{+53}$ | 153 | 0.9 , 1.2 | 86.39 | 51.34 | **85.83** | 55.13 |

In these experiments, we use $\lambda = 3.5$, which is the default setting for the NuAT-WA 50-epoch results reported in Table-1(b) of the main paper. We use the same ramp-up rate for $\lambda$ (as the 50-epoch training) during the 20 epochs of learning rate ramp-up across all experiments in Table-2 in order to avoid stability issues during this period. Beyond this, the value of $\lambda$ increases linearly to 3.5 over the remaining epochs. As we move to longer training schedules, we increase $t_{min}$ and $t_{max}$ as shown in Table-2. This increases the effective number of training epochs as well, which is also reported in the table. As we increase the number of training epochs, we obtain better results upto 80 epochs of training. We obtain significantly improved results for the 80-epoch training which uses 112 epochs effectively (considering the overhead due to two-step attacks in the hybrid approach). The accuracy on the GAMA-PGD attack is 55.35% and accuracy on the AutoAttack is 54.97%, which is 1.61% higher than the NuAT-WA results reported in Table-1(b) of the main paper. In fact, this is 0.21% higher than even the NuAT2-WA results, which is a two-step defense run for 80 training epochs, with an effective number of training epochs as 160. Therefore, using the proposed hybrid training algorithm, we achieve better results at a significantly lower computational cost. We note that the clean accuracy of NuAT2-WA is 0.92% higher than the hybrid approach.

## 3  Details On Datasets Used

We run extensive evaluation on the following three benchmark datasets: MNIST [8], CIFAR-10 [7] and a 100 class subset of ImageNet [4, 12]. For each dataset, we maintain a train-validation split that is balanced equally across all classes.

**MNIST** is a 10-class dataset consisting of grayscale handwritten images of size $28 \times 28$. The dataset consists of $60,000$ training images, and $10,000$ test images. We further partition the original training dataset into $50,000$ images for training, and a $10,000$ image hold-out set for validation.

**CIFAR-10** consists of colour images of size $32 \times 32 \times 3$, and is a commonly used dataset for benchmarking adversarial attacks and defenses. The original dataset consists of $50,000$ training images and $10,000$ test images. For CIFAR-10, we maintain a validation split of $1,000$ images, and use $49,000$ images for training.

**ImageNet-100:** Further, we present results on a random 100-class subset of ImageNet to validate the proposed approach. Given the large input size of images ($224 \times 224 \times 3$), and the high degree of similarity between classes, achieving robust models on this dataset is a difficult task, and serves as a representative for challenges faced in real-world deployment. Since the original ImageNet dataset has a private test set, we use the designated validation set as the test set. Further we take the designated training set and partition it so as to maintain an $80 - 20$ train-validation split. We share the list of the 100 classes used along with our codes.

## 4  Details on Training Methodology

In this section, we present details on the training methodology of the proposed defense. We use Nvidia RTX 2080 TI and Nvidia-V100 GPUs for our experiments. We share our code along with the Supplementary submission.

Table 3: **Computational Complexity** of various robust defenses trained on the CIFAR-10 dataset using the WideResNet-34-10 architecture. Training time per epoch in seconds, and total training time in hours are presented along with robust accuracy (%) obtained against AutoAttack.

| Training Method | Time per Epoch (sec) | Number of Epochs | Total Time (hours) | Accuracy (AA in %) |
|---|---|---|---|---|
| FBF | 190 | 30 | 1.58 | 43.10 |
| GAT | 328 | 85 | 7.74 | 50.12 |
| GAT-WA | 330 | 85 | 7.79 | 51.50 |
| NuAT | 352 | 55 | 5.38 | 50.81 |
| NuAT-WA | 353 | 50 | 4.90 | 53.36 |
| NuAT-H25 | 365 | 25 | 2.53 | 51.14 |
| NuAT-H50 | 368 | 50 | 5.10 | 51.58 |
| NuAT-H60 | 368 | 60 | 6.12 | 53.81 |
| NuAT-H80 | 465 | 80 | 10.33 | 54.85 |
| NuAT2 | 499 | 55 | 7.62 | 51.27 |
| NuAT2-WA | 500 | 80 | 11.11 | 54.76 |
| TRADES | 1470 | 110 | 44.92 | 53.80 |
| PGD | 1064 | 200 | 59.11 | 52.19 |
| AWP | 1469 | 200 | 81.61 | 56.17 |

## 4.1 Implementation Details

**Architecture:** On MNIST, we present results using a modified LeNet [9] architecture using two additional convolutional layers, similar to Sriramanan et al. [12]. For the CIFAR-10 and ImageNet-100 datasets, we present results using the ResNet-18 [6] architecture. We additionally present results on CIFAR-10 using the WideResNet-34-10 [16] architecture for the proposed defense. We also present additional results on ImageNet-100 using the ResNet-34 architecture in Sec-6. This helps highlight the efficacy of exponential model weight averaging in models of larger capacity particularly on challenging datasets such as ImageNet-100.

**Number of Training epochs:** We train the ResNet-18 models of CIFAR-10 and ImageNet-100 for 100 and 110 epochs respectively. MNIST is trained for 20 epochs on the Modified-LeNet architecture. We additionally show results with lower number of training epochs on WideResNet-34-10 architecture for CIFAR-10. On this architecture, for the NuAT, NuAT-2, NuAT-WA and NuAT2-WA training we use 55, 55, 50 and 80 epochs respectively. While it is possible to fix the number of epochs to 50 for all cases, we report results with the optimal trade-off between accuracy and computational complexity. We further compare the computational complexity in tabular form in Table-3, which highlights the efficacy of the proposed approach.

## 4.2 Learning Rate Schedule

**Cyclic-Step Learning Rate Schedule:** We use Stochastic Gradient Descent (SGD) optimizer with a novel Cyclic-Step learning rate schedule for our CIFAR-10 experiments. We find that cyclic learning rate schedule helps achieve a better accuracy-robustness trade-off in the initial phase of training, whereas a step learning rate schedule helps achieve a significant boost in both robustness and clean accuracy towards the end, as observed by Rice et al. [11]. We present the learning rate schedule for the 100-epoch training regime in Fig.1. Here, the learning rate increases linearly from 0 to 0.1 in 40 epochs, and decreases from 0.1 to 0 in 60 epochs. However, after epoch 80, we switch over to a step learning rate schedule with a learning rate drop of 10 after epoch 85, and a second drop of 10 after epoch 90. For the CIFAR-10 ResNet-18 model, we use a variant of the schedule illustrated in Fig.1, by decreasing the learning rate from 0.1 to 0 in 50 epochs (rather than 60).

**Cyclic Learning Rate Schedule:** While the proposed Cyclic-Step learning rate schedule results in better performance on the base NuAT model, cyclic learning rate schedule results in better clean and robust accuracy after weight averaging as shown in Table-4. This indicates that a smooth schedule which yields multiple similar models towards the end of training leads to a higher boost after weight-

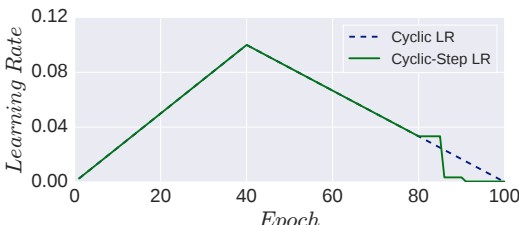

Figure 1: **Cyclic and Cyclic-Step Learning Rate schedules:** Learning rate increases linearly from 0 to 0.1 in 40 epochs and decreases back to 0 over the next 60 epochs in Cyclic Learning Rate schedule. The proposed Cyclic-Step Learning Rate schedule transitions from Cyclic LR to Step LR after epoch-80 with a drop by a factor of 10 after epoch 85 and 90.

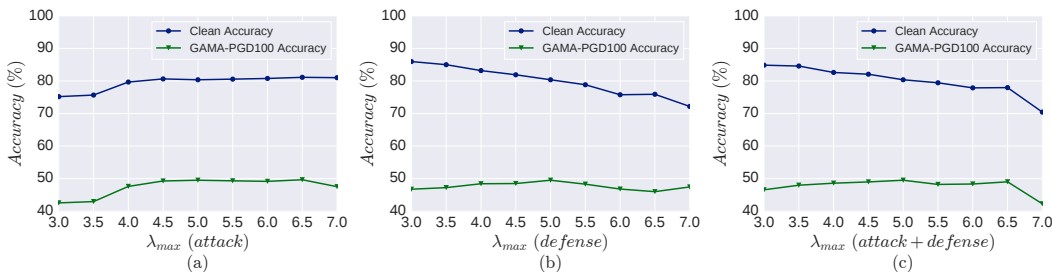

Figure 2: **Sensitivity across variation in $\lambda$:** Accuracy on clean samples and robust accuracy against GAMA-PGD 100-step attack are plotted against variation in $\lambda_{max}$. (a) $\lambda_{max}$ for the defense is set to 5 and $\lambda_{max}$ for attack is varied. (b) $\lambda_{max}$ for the attack is set to 5 and $\lambda_{max}$ for defense is varied. (c) $\lambda_{max}$ is varied for both attack and defense.

averaging when compared to a sudden increase in accuracy obtained using step learning rate decay. For the ResNet-18 training on CIFAR-10 dataset we ramp up the learning rate from 0 to 0.1 in the first 25 epochs followed by a ramp down over the remaining 75 epochs.

The learning rate schedules for different training regimes are detailed in the ReadMe file along with our code submission.

### 4.3 Sensitivity across variation in $\lambda$

The proposed Nuclear Norm regularizer is weighted by a factor $\lambda$, (refer Eq.2 of the main paper) which enforces smooth function values locally by using the joint batch-statistics of adversarial samples. By altering this weighting factor used during training, we can thereby control the robustness-accuracy trade-off so achieved between the extremes of a normal model and an overly-smooth model that achieves low standard accuracy. We use the same setting of $\lambda$ for both adversary generation (Eq.1 in the main paper) and training (Eq.2 in the main paper). We increase the weight of the Nuclear-Norm term linearly from 0 to $\lambda_{max}$ over the training epochs. This helps achieve a significantly better accuracy-robustness trade-off when compared to using a fixed value of $\lambda$. We show the robustness-accuracy trade-off against variation in $\lambda_{max}$ on the CIFAR-10 dataset in Fig.2 (c). We note that for

Table 4: **Ablation experiments** to compare the performance across different **learning rates schedules** in a ResNet-18 model on CIFAR-10 dataset. Prediction accuracy (%) on the original dataset (Clean) and robust accuracy (%) against GAMA PGD 100-step white-box attack are presented.

| Learning Rate | Clean (Base-model) | GAMA-100 (Base-model) | Clean (WA model) | GAMA-100 (WA model) |
|---|---|---|---|---|
| Cyclic LR | **83.24** | 48.70 | **82.21** | **50.97** |
| CycStep LR | 81.01 | **49.46** | 79.95 | 50.63 |

Table 5: **Statistical summary** from multiple reruns with random seeds, to highlight the stability of training for different variants of the proposed defense. Prediction accuracy (%) on the original dataset (Clean) and robust accuracy (%) against GAMA PGD100-step white-box attack are presented on the CIFAR-10 dataset.

| Training Method | Clean | | GAMA-100 | |
|---|---|---|---|---|
| | Mean | Std-Dev | Mean | Std-Dev |
| NuAT | 81.57 | 0.40 | 49.30 | 0.29 |
| NuAT-H-WA | 81.97 | 0.11 | 50.77 | 0.18 |
| NuAT2-WA | 81.70 | 0.13 | 50.61 | 0.13 |

lower values of $\lambda_{max}$, clean accuracy is higher, whereas the accuracy against GAMA-100 attack is lower. As the value of $\lambda_{max}$ increases, the clean accuracy reduces and robustness improves. We observe a catastrophic failure of model training as discussed by Wong et al. [15] for very high values of $\lambda_{max}$ ($\geq 7$). We use a $\lambda_{max}$ of 4.5 for CIFAR-10 on ResNet-18 architecture and 4 for WideResNet-34-10. For MNIST we use $\lambda_{max}$ of 1, and for ImageNet-100 we use the setting of 1.5 for $\lambda_{max}$. For the CIFAR-10 ResNet-18 NuAT model, we find that setting the value of $\lambda$ to be constant in the last 10 epochs improves the clean accuracy without sacrificing robust accuracy.

We use a lower value of of $\lambda$ for the weight-averaging case since it results in better clean accuracy at the cost of sub-optimal robustness on the base model. However the robust accuracy is boosted after weight averaging, with a small drop in the clean accuracy, leading to an significantly improved clean and robust accuracy when compared to the NuAT model. The settings used for training NuAT-WA and NuAT2-WA models are detailed in the ReadMe file along with the code submission.

Although we use the same value of $\lambda$ for both adversary generation and training, we analyse the impact of varying both independently as well. In Fig.2(a) we observe the clean accuracy and robustness for the case of $\lambda_{max,defense}$ fixed to the setting of 5 and $\lambda_{max,attack}$ varying from 3 to 7. For very low values of $\lambda_{max,attack}$, we observe catastrophic overfitting, whereas the results are more stable for higher values. This indicates that the use of Nuclear-Norm term for attack generation helps mitigate gradient masking effects, thereby preventing the generation of weak adversaries, leading to more stable training. The setting of $\lambda_{max,attack} = 6.5$ results in clean accuracy of $81.11\%$ and GAMA-PGD accuracy of $49.66\%$, which is better than the final results we report, using a common setting for both attack and training. The results for a fixed value of $\lambda_{max,attack}$ and variation in $\lambda_{max,defense}$ are reported in Fig.2(b). The trend of clean accuracy and robustness for this case is similar to that of Fig.2(c), however the results degrade earlier for higher values of $\lambda_{max,defense}$, due to the lower value of $\lambda_{max,attack}$ compared to Fig.2(c).

## 4.4 Stability of Nuclear Norm Adversarial Training

To analyse the stability of different variants of the proposed defense, we train ResNet-18 models on the CIFAR-10 dataset multiple times using random seeds for initialisation of network parameters. For consistency, we train the NuAT, NuAT-H-WA and NuAT2-WA models using the methodology as laid out in Sections-2,4. In Table-5, we present the Mean and Standard Deviation of both clean and robust accuracy, with the latter evaluated against the GAMA-PGD 100-step adversarial attack. We observe that all variants of the proposed defense are remarkably stable across reruns. In particular, we highlight the efficacy of Hyrbid-NuAT in providing enhanced performance and stability similar to that of *effective* multi-step defenses, while requiring only a nominal computational overhead (below 3%) as compared to *efficient* single-step adversarial training. In stark contrast, if an RFGSM-AT [13] model is trained 10 times with random initialisations, we observe zero robust accuracy against GAMA PGD 100-step attack in the last epoch for 5 runs, indicating catastrophic failure for half the experimental reruns.

Table 6: **Ablation experiments** to compare the performance achieved by a ResNet-18 model on the CIFAR-10 dataset by using different **attack methods** and **training losses**. Prediction accuracy (%) on the original dataset (Clean) and robust accuracy (%) against GAMA PGD 100-step white-box attack are presented.

| Ablations | Clean | PGD-100 | GAMA-100 |
|---|---|---|---|
| Nu-AT (Proposed method) | 81.01 | 52.93 | 49.46 |
| **(A1):** Nu-AT using only CE attack | 83.75 | 49.58 | 47.21 |
| **(A2):** Nu-AT using only CE+NN attack | 82.46 | 51.90 | 48.37 |
| **(A3):** RFGSM training with CE+NN attack | 90.67 | 31.72 | 31.25 |

## 5 Ablation Analysis of Proposed Approach

### 5.1 Ablations on Attack and Training Loss

We perform a detailed ablation study to determine the contribution of the attack utilised during adversarial training on the extent of robustness achieved, as shown in Table-6. We present these ablations using ResNet-18 models trained on the CIFAR-10 dataset. First, we experiment with using the proposed loss (Eq.2 of the main paper), consisting of the cross-entropy loss and the Nuclear Norm regularizer, to train on R-FGSM adversaries generated by maximising only the cross-entropy loss (A1). Next, we perform Nuclear Norm adversarial training on adversaries wherein the combined loss is used in every minibatch throughout the entire training regime (A2), that is, without alternating with R-FGSM adversaries in alternate minibatches as described in Section-5 of the main paper. We find that while training on adversaries generated using the combined loss leads to an improvement over training on standard R-FGSM adversaries, utilizing each attack in alternate minibatches leads to sizeable improvement in robust accuracy.

To highlight the efficacy of the proposed regulariser in adversarial training, we next train a model by minimizing the cross-entropy on adversaries generated using the combined loss (A3). Similar to standard R-FGSM training, we observe that the associated training dynamics is highly volatile and is susceptible to catastrophic failure, and thus obtains significantly poorer robust performance.

### 5.2 Ablations on Weight-Averaging

The NuAT-WA and NuAT2-WA algorithms utilize exponential weight averaging as discussed in the main paper and Algorithm-1. We perform ablation experiments to identify the most important layers which contribute to a boost in accuracy using weight averaging on the NuAT2-WA defense in Table-7. We use the WideResNet-34-10 architecture on CIFAR-10 dataset for these experiments. We consider the base model as the one trained using SGD, and for each of the rows in the table, we adapt the weights of the respective layers (or blocks) from the weight averaged model. We incrementally add layers starting from the output layer in each of the rows. While addition of the last linear layer results in a small boost in the clean and robust accuracy, the most significant boost ($52.97\%$ to $57.22\%$) is achieved by using the weights of Block-3 from the weight averaged model. This shows that smoothing of the final layers of the CNN is most significant for the gains achieved using weight averaging. Addition of Block-2 does not cause a change in either clean or robust accuracy, while addition of the first convolutional layer and Block-1 leads to a significant boost in clean accuracy.

## 6 Detailed Results on ImageNet-100

While we achieve significant improvement in results on the WideResNet-34-10 architecture for CIFAR-10 using weight averaging ($2.55\%$ boost in GAMA accuracy from Table-1(b) of the main paper), the improvement on ResNet-18 architecture with weight averaging is lower ($1.51\%$ from Table-1(a) of the main paper). We note that as expected, the gains in robustness are higher for large capacity models. We also observe higher gains in robustness on ImageNet-100 dataset on ResNet-34 architecture as shown in Table-8. For 1-step defense, the robust accuracy improves from $36.24\%$ to $37.28\%$ using weight-averaging, while for the 2-step defense, the robust accuracy improves from $37.9\%$ to $38.92\%$. We also note that the use of 2-step defense boosts the accuracy by around $1.64\%$ on ResNet-34 architecture, while the gains on ResNet-18 are marginal.

Table 7: **Ablation experiments** to highlight the importance of weight averaging of specific layers in a WideResNet-34-10 model on CIFAR-10 dataset. Prediction accuracy (%) on the original dataset (Clean) and robust accuracy (%) against PGD 20-step white-box attack are presented.

| Weights adapted from NuAT2-WA model | Clean | PGD-20 |
|---|---|---|
| none | 87.29 | 52.10 |
| Linear | **87.33** | 52.97 |
| Linear, Block-3 | 84.51 | 57.22 |
| Linear, Block-3, Block-2 | 84.18 | 57.62 |
| Linear, Block-3, Block-2, Block-1 | 85.48 | 57.60 |
| Linear, Block-3, Block-2, Block-1, Conv-1 (all) | 86.32 | **58.05** |

Table 8: **ImageNet-100 white-box results:** Accuracy (%) of the proposed defense across different model architectures. We obtain significant gains in accuracy using NuAT-WA and NuAT2-WA on larger models.

| Method | Architecture | Clean | PGD-20 | GAMA-100 |
|---|---|---|---|---|
| NuAT | ResNet-18 | 69.00 | 38.48 | 32.38 |
| NuAT-WA | ResNet-18 | 68.40 | 39.56 | 33.22 |
| NuAT | ResNet-34 | 73.16 | 41.54 | 36.24 |
| NuAT-WA | ResNet-34 | **73.22** | 42.62 | 37.28 |
| NuAT2 | ResNet-34 | 71.64 | 43.42 | 37.90 |
| NuAT2-WA | ResNet-34 | 71.04 | **44.32** | **38.92** |

# 7    Scalability to ImageNet-1k

To study the efficacy of the proposed approach on large-scale datasets such as ImageNet, we integrate our code with the FBF [15] repository, since this has several optimizations such as half-precision computations. We report results of the FBF baseline and NuAT (Ours) using the ResNet-50 architecture on images resized to 352×352 and cropped to 288×288 in Table-9. Similar to FBF, we consider robustness within the $\ell_\infty$ constraint of 4/255. While previously reported results are based on evaluations on the PGD 50-step attack, we additionally report results against GAMA-PGD 100-step attack to obtain a more reliable estimate of the true robustness. Using NuAT, we achieve 3.32% boost in clean accuracy and 7.54% boost in robustness against the GAMA-PGD 100-step attack. By using Weight averaging (NuAT-WA), we obtain a further boost in robust performance. This demonstrates that the proposed approach indeed scales to ImageNet-1k.

# 8    Additional Results against White-Box Attacks

We evaluate the proposed NuAT and NuAT-WA defenses on all three datasets by using targeted and untargeted 1000-step PGD adversaries, as shown in Table-10 and Table-11 respectively. Among the targeted attacks, we consider 1000-step attack with the Least Likely class as the target, and another 1000-step attack targeted toward a random class. We observe that the PGD attack indeed converges, since the accuracy against both 500-step and 1000-step adversaries are nearly identical for both NuAT and NuAT-WA across the three datasets. Next, we present the worst-case robustness of the proposed

Table 9: **ImageNet-1K evaluation:** Prediction accuracy (%) on the original dataset (Clean) and robust accuracy (%) against two white-box attacks are presented using the ResNet-50 architecture, under the $\ell_\infty$ constraint set of $\varepsilon =$4/255.

| Training Method | Clean | PGD-50 | GAMA-PGD |
|---|---|---|---|
| FBF | 56.70 | 31.96 | 27.44 |
| NuAT (**Ours**) | **60.02** | 41.56 | 34.98 |
| NuAT-WA (**Ours**) | 59.65 | **41.87** | **35.42** |

Table 10: Prediction accuracy (%) of **NuAT in various targeted and untargeted White-Box attack settings**. In the second partition, we present worst-case performance across multiple random restarts of the PGD attack. We consider a 1000-sample subset of CIFAR-10 and ImageNet-100 datasets for the random restarts experiments.

| Attack | CIFAR-10 | | ImageNet-100 | | MNIST | |
|---|---|---|---|---|---|---|
| | 500-step | 1000-step | 500-step | 1000-step | 500-step | 1000-step |
| PGD-Targeted Attack (Least Likely class) | 77.20 | 77.20 | 65.70 | 65.50 | 98.99 | 98.99 |
| PGD-Targeted Attack (Random class) | 73.70 | 73.60 | 64.90 | 64.00 | 98.81 | 98.72 |
| PGD-Untargeted Attack | 52.97 | 52.97 | 37.60 | 37.60 | 96.46 | 96.46 |
| | 1-RR | 1000-RR | 1-RR | 1000-RR | 1-RR | 1000-RR |
| PGD 50-step, 1000-RR | 51.30 | 50.60 | 37.80 | 37.30 | 96.71 | 94.42 |

Table 11: Prediction accuracy (%) of **NuAT-WA in various targeted and untargeted White-Box attack** settings. In the second partition, we present worst-case performance across multiple random restarts of the PGD attack. We consider a 1000-sample subset of CIFAR-10 and ImageNet-100 datasets for the random restarts experiments.

| Attack | CIFAR-10 | | ImageNet-100 | | MNIST | |
|---|---|---|---|---|---|---|
| | 500-step | 1000-step | 500-step | 1000-step | 500-step | 1000-step |
| PGD-Targeted Attack (Least Likely class) | 79.50 | 79.50 | 64.50 | 64.10 | 98.92 | 98.92 |
| PGD-Targeted Attack (Random class) | 75.50 | 75.50 | 63.50 | 63.00 | 98.77 | 98.75 |
| PGD-Untargeted Attack | 53.95 | 53.95 | 38.70 | 38.60 | 96.33 | 96.33 |
| | 1-RR | 1000-RR | 1-RR | 1000-RR | 1-RR | 1000-RR |
| PGD 50-step, 1000-RR | 54.20 | 53.20 | 39.40 | 38.20 | 96.70 | 94.45 |

approach against 50-step PGD adversaries run over multiple restarts in the lower partition of the Tables-10 and 11. Since this attack evaluation is computationally intensive, we utilize a 1000-sample balanced subset for the CIFAR-10 and ImageNet-100 datasets. We observe that the fall in accuracy is minimal, and indeed saturates to a stable value as is desired. We also note that the saturating accuracy in all cases is still higher than that obtained on the GAMA attack and AutoAttack as detailed in Table-2 of the main paper.

We use the early version (v1) of AutoAttack for reporting results in Table-1(a) of the main paper since we utilize the baseline results from Sriramanan et al. [12]. However, for the WideResNet-34-10 evaluations in Table-1(b) of the main paper, we use the latest version (v2) of AutoAttack.

# 9 Black-Box and Gradient-Free Attacks

We present transfer-based black-box attacks (FGSM and PGD 7-step) on the proposed NuAT and NuAT-WA defenses in Table-12 and Table-13 respectively. We note that across all datasets, black-box transfer attacks are weaker when compared to the strong white-box attacks (GAMA-PGD and AutoAttack) in as detailed in Table-2 of the main paper, indicating the absence of gradient masking. We observe that the transfer-based black-box accuracies closely track the clean accuracy of the target model.

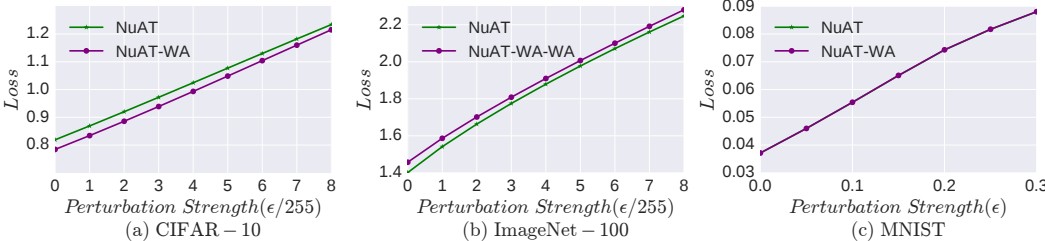

(a) $CIFAR-10$     (b) $ImageNet-100$     (c) MNIST

Figure 3: **CE loss on FGSM attack vs. Perturbation strength:** The cross-entropy loss on perturbations generated using FGSM attack plotted against variation in the $\ell_\infty$ perturbation bound for the proposed defenses NuAT and NuAT-WA. The loss increases monotonically with an increase in $\varepsilon$, indicating the absence of gradient masking in the proposed single-step defense [2].

Table 12: **Accuracy of NuAT models against Black-Box and Gradient-Free Attacks:** Accuracy (%) of the proposed NuAT defense against transfer-based black-box attacks (FGSM and PGD 7-step), and the gradient-free attack, Square [1].

| Attack | CIFAR-10 | | ImageNet-100 | | MNIST | |
|---|---|---|---|---|---|---|
| | Source | Acc (%) | Source | Acc (%) | Source | Acc (%) |
| Clean | NA | 81.01 | NA | 69.00 | NA | 99.37 |
| FGSM | VGG11 | 77.41 | AlexNet | 67.34 | BB-MNIST | 96.50 |
| FGSM | ResNet18 | 79.64 | ResNet18 | 67.30 | M-LeNet | 96.41 |
| PGD 7-step | VGG11 | 78.26 | AlexNet | 68.08 | BB-MNIST | 96.97 |
| PGD 7-step | ResNet18 | 79.52 | ResNet18 | 68.22 | M-LeNet | 97.11 |
| Square | NA | 55.41 | NA | 42.56 | NA | 93.22 |

Table 13: **Accuracy of NuAT-WA models against Black-Box and Gradient-Free Attacks:** Accuracy (%) of the proposed NuAT-wA defense against transfer-based black-box attacks (FGSM and PGD 7-step), and the gradient-free attack, Square [1].

| Attack | CIFAR-10 | | ImageNet-100 | | MNIST | |
|---|---|---|---|---|---|---|
| | Source | Acc (%) | Source | Acc (%) | Source | Acc (%) |
| Clean | NA | 82.21 | NA | 68.40 | NA | 99.36 |
| FGSM | VGG11 | 78.92 | AlexNet | 66.04 | BB-MNIST | 96.49 |
| FGSM | ResNet18 | 80.18 | ResNet18 | 66.30 | M-LeNet | 96.41 |
| PGD n-Step | VGG11 | 79.66 | AlexNet | 67.12 | BB-MNIST | 96.89 |
| PGD n-Step | ResNet18 | 80.70 | ResNet18 | 67.00 | M-LeNet | 97.07 |
| Square | NA | 56.54 | NA | 42.14 | NA | 93.30 |

To further verify the absence of gradient masking in the proposed NuAT defense, we perform evaluations on gradient-free adversaries such as the SPSA attack (Simultaneous Perturbation Stochastic Approximation) [14] and Square attack [1]. SPSA uses a numerical approximation to the gradient by computing the function values along randomly sampled directions. Here, we use 100 random directions to approximate the gradient using SPSA. On the CIFAR-10 dataset, a ResNet-18 trained using the proposed defense obtains 60.1% on the SPSA attack. We note that white-box attacks (Table-1 of the main paper) are far stronger due to the presence of reliable gradients, indicating the absence of gradient masking.

We also evaluate the NuAT and NuAT-WA defenses on the Square Attack in Table-12 and Table-13, which uses zeroth-order optimisation to compute adversarial samples. We obtain a robust accuracy of 55.34% for a NuAT trained ResNet-18 model. Since the Square Attack is significantly stronger and more computationally efficient as compared to the SPSA attack, we only present results on the former attack on the ImageNet-100 and MNIST datasets. We observe that the Square attack is weaker when

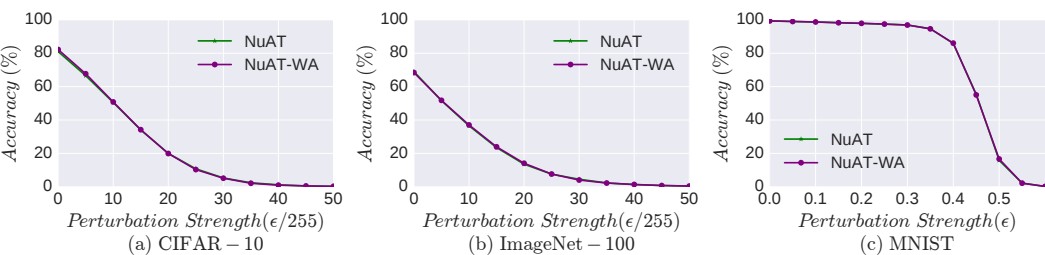

Figure 4: **Robust Accuracy (%) against PGD $n$-step attack across varying perturbation strengths:** The robust accuracy of the proposed defense NuAT is plotted against variation in $\ell_\infty$ perturbation bound for the proposed defenses NuAT and NuAT-WA. The accuracy decreases to 0 as perturbation strength ($\varepsilon$) increases, indicating the absence of gradient masking in the proposed single-step defense [2]. We use a 7-step attack for CIFAR-10 and ImageNet-100 datasets and 40-step attack for MNIST.

Table 14: **Adaptive Attacks:** Accuracy (%) of the proposed defense Nu-AT against adaptive attacks on CIFAR-10 dataset. CE: cross-entropy loss, NN: Nuclear-Norm regularizer, MM: Maximum-margin loss

| $\lambda$ | 20-step attack | | 100-step attack | |
|---|---|---|---|---|
| | **CE+NN** | **MM+NN** | **CE+NN** | **MM+NN** |
| -5 | 79.93 | 79.39 | 52.45 | 50.01 |
| -3 | 79.52 | 78.58 | 52.46 | 50.00 |
| -2 | 79.30 | 76.19 | 52.41 | 49.99 |
| -1 | 76.30 | 54.60 | **52.39** | 50.20 |
| 0 | **53.16** | 51.17 | 52.45 | 50.60 |
| 1 | 54.50 | **50.92** | 52.40 | 50.06 |
| 2 | 57.01 | 52.33 | 52.43 | **49.97** |
| 3 | 59.37 | 53.70 | 52.53 | 50.14 |
| 5 | 61.91 | 57.26 | 52.52 | 50.25 |
| 10 | 64.28 | 61.82 | 52.43 | 50.62 |
| 20 | 65.36 | 64.26 | 52.46 | 50.68 |
| 30 | 65.90 | 65.01 | 52.49 | 50.84 |

compared to white-box attacks (GAMA-PGD and AutoAttack) on CIFAR-10 and ImageNet-100, whereas it stronger than GAMA-PGD on MNIST. The proposed defense NuAT however achieves improved robustness compared to all other defense methods, including multi-step methods such as TRADES and PGD-AT, on the AutoAttack ensemble which includes the Square attack (Table-2 of main paper).

## 10 Details on Adaptive Attacks

We evaluate the proposed defense against Adaptive Attacks which are cognizant of the defense algorithm [3]. We use the following loss, which is similar to the adversary generation and training for the proposed defense:

$$L = \ell\left(f_\theta(X + \Delta), Y\right) + \lambda \cdot ||f_\theta(X + \Delta) - f_\theta(X)||_* \tag{1}$$

We present results against attacks generated by finding the optimal $\Delta$ using PGD 20-step and 100-step optimization respectively in Table-14. Similar to the GAMA attack [12], we decay the value of $\lambda$ over the first 25 iterations in the 100-step adaptive attacks. We note that this generates stronger attacks compared to the 20-step attacks. We consider both cross-entropy and maximum-margin loss in the first term of the above equation for adaptive attacks, and observe that the latter generates stronger attacks. We present results by varying the value of $\lambda$, and note that the range of $\lambda$ from $-2$ to $2$ produces the strongest attacks. We note that the adaptive attacks presented in Table-14 are not stronger than the GAMA-PGD attack. Therefore, evaluation of the proposed defense against GAMA-PGD attack and AutoAttack is sufficient to obtain a reliable estimate of its robustness.

## 11 Details on Gradient Masking checks

The plot of average cross-entropy loss on FGSM adversaries against variation in attack perturbation bound ($\varepsilon$) for the NuAT and NuAT-WA defenses is presented in Fig.3. We note that within the local vicinity of the data samples, loss increases monotonically with increase in $\varepsilon$. This shows that the loss surface is smooth, thereby indicating the absence of gradient masking in the proposed single-step defenses NuAT and NuAT-WA [2].

The plot of robust accuracy against multi-step PGD attacks across variation in attack perturbation bound ($\varepsilon$) is shown in Fig.4. For this attack, we use 7 steps for CIFAR-10 and ImageNet-100 datasets, and 40 steps for MNIST. As the $\ell_\infty$ bound on the perturbations increases, the attack becomes stronger and the robust accuracy decreases to 0 in an identical manner for both NuAT and NuAT-WA models.

This shows that gradient-based attacks are sufficiently strong, thereby indicating the absence of gradient masking in the proposed single-step defenses [2].