# OpenReview forum: "Towards Efficient and Effective Adversarial Training"
_NeurIPS.cc/2021/Conference — NeurIPS 2021 Poster_

### Official Review · Reviewer_MFYP · 2021-07-12

**Rating:** 7
**Confidence:** 4

**Summary:**

This paper proposes a new adversarial training method using the Nuclear regularizer with weight averaging to bridge the gap of the trained robustness accuracy between single-step and multi-step adversarial training. Comprehensive experimental results show that the proposed method can successfully eliminate the problem of gradient masking for single-step adversarial training methods while can still achieve high robustness performance as multi-step ones.

**Limitations And Societal Impact:**

No limitations and potential negative societal impact.

**Main Review:**

Strength:
- The proposed method is novel and the investigated problem is challenging. Using nuclear regularization seems to be an interesting solution. Insights behind the proposed methods are well explained.
- The authors have compared the proposed methods with most SOTA adversarial training methods and strong white-box and black-box attacks. I love the comprehensive and informative evaluations presented in both submission draft and supplementary.

Weakness:
- The proposed method is very simple and straightforward.
- There is no theoretical guarantee that the proposed method can always eliminate the problem of gradient masking.

Comments:
- Can the proposed 1-step and 2-step NuAT always prevent gradient masking? I understand providing a theoretical guarantee might be difficult, but multiple times of training tests would be sufficient. For instance, how many times the gradient masking will happen if we train with the proposed methods (1-step and 2-step NuAT independently) using over 10 different random initial weights compared to RFSGM-AT for long training epochs (e.g., 100 epochs)?
- The training process is usually full of randomness. It would be better to do the training multiple times (at least 3 times) and show the mean and variance in the main table (e.g., Table 1a). That would make the results more convincing.
- It would be helpful to provide the detailed numbers of training time and epoch time to get a better understanding of the computational complexity of the proposed methods.

--- Post Rebuttal ---
Thank the authors for the detailed response and additional results. Most of my concerns are addressed. I will raise my score and vote for acceptance.

**Time Spent Reviewing:**

3.5

---

> ### Author Response · Authors · 2021-08-11
> **Response to reviewer MFYP**
>
> We sincerely thank the reviewers for their time and valuable feedback on our work. We are happy to see that the reviewers find our work novel and significant, our evaluations comprehensive and thorough, and our results noteworthy in terms of effectiveness and efficiency. It is indeed motivating to find that all reviewers find our work to be worthy of acceptance at NeurIPS. We will certainly work towards incorporating all the suggestions in our camera-ready version.
>
> We address the specific comments posted by the reviewer *MFYP* below:
>
>  -  **Susceptibility to gradient masking**
>
>      -  The proposed approach does not theoretically guarantee that the problem of gradient masking can always be eliminated. As pointed out, convergence guarantees are extremely difficult to provide in a practical setting (Eg: for a ResNet-18 network on CIFAR-10), especially in the challenging domain of adversarial robustness for real-world datasets. However, once a final model is obtained, certificates can be provided to its robustness by combining the proposed approach with techniques such as randomized smoothing [A, B].
>
>     -   As suggested, we perform 10 reruns with random initialization of weights for RFGSM-AT and single-step and two-step variants of NuAT. Out of 10 runs, the RFGSM-AT defense obtains zero robust accuracy (against GAMA PGD 100-step attack [25]) in the last epoch for 5 runs, indicating catastrophic failure for 5 out of 10 runs. Using all variants of the NuAT defense, we are able to train robust models successfully in all 10 runs with no such catastrophic failures.
>
>     -  As noted in L331-333, the proposed NuAT single-step defense is significantly more stable compared to RFGSM-AT. However, it is not as stable as the two-step defense for large model capacities and longer training schedules. The hybrid approach (NuAT-H) effectively mediates between the 1-step and 2-step approaches by achieving results similar to the 2-step defense at a very small computational overhead. Using NuAT-H-WA, we obtain results as good as NuAT2-WA (as seen in the table below) at a significantly lower computational cost. The additional training overhead for all 10 reruns of NuAT-H-WA is within 3 epochs, which is below 3%.
>
>
>      [A] Cohen et al., Certified Adversarial Robustness via Randomized Smoothing, ICML 2019
>
>      [B] Salman et al., Provably Robust Deep Learning via Adversarially Trained Smoothed Classifiers, NeurIPS 2019
>
>
> -  **Mean and variance across reruns**
>
>      The mean and standard deviation on the CIFAR-10 dataset and ResNet-18 architecture are shown below:
>
>      Robust accuracy is computed against the GAMA PGD 100-step attack [25]. We will include this in the final version as suggested.
>
>     |           | Clean (Average) | Clean (Std. Dev) | Robust (Average) | Robust (Std. Dev) |
> |-----------|:---------------:|:----------------:|:----------------:|:-----------------:|
> |    NuAT   | 81.57           | 0.40             | 49.30            | 0.29              |
> | NuAT-H-WA | 81.97           | 0.11             | 50.77            | 0.18              |
> |  NuAT2-WA | 81.70           | 0.13             | 50.61            | 0.13              |
>
> -   **Detailed numbers of training time and epoch time**
>
>      We will include the plot in Fig.3(a) as a table in the final version of the submission.

---

### Official Review · Reviewer_5nmX · 2021-07-14

**Rating:** 7
**Confidence:** 4

**Summary:**

The paper addresses the performance gap between single-step and multi-step adversarial training by introducing a Nuclear Norm regularizer to improve the adversarial robustness of models in single-step adversarial training. It shows that this regularizer encourages function smoothing near clean samples by incorporating joint batch-statistics of adversarial samples, which leads to increased robustness. The authors also use exponential weight-averaging and a 2-step variant of the proposed defense to further improve the performance. Experiments on MNIST, CIFAR-10 and ImageNet-100 demonstrate that the proposed method outperforms single-step adversarial training methods and is competitive with multi-step methods in terms of robustness (while being more computationally efficient).

**Limitations And Societal Impact:**

Yes

**Main Review:**

- The paper is well-written and the proposed approach is described clearly.


- Experiments are thorough and demonstrate that the method outperforms most single-step adversarial training approaches and is competitive with multi-step methods in terms of robustness, while being more computationally efficient than multi-step methods.


- The authors verify how the proposed regularizer encourages local smoothness of the loss surface and more diversity in the generated attacks.


- As common in adversarial training methods, the proposed approach leads to a drop in clean accuracy. I am wondering if the proposed method can be combined with methods such as AdvProp [B] which alleviate the clean accuracy drop in adversarial training.
Specifically, whether using separate batch norms for clean and adversarial examples is possible in the proposed framework and whether it can lead to improvements in clean accuracy.


- Comparison with AWP and GAT-WA is missing in Table 2. The authors mention that "though we expect AWP training to be highly effective, we do not present the method here due to the complexity of finding hyperparameters and the immense computational cost incurred for each training run, particularly in the case of ImageNet-100" (L 306 - 308). But it would be good to have these results in the final version for completeness.
There are also other adversarial training defenses such as [A] that are not mentioned in the paper.


- The authors claim that "the proposed method NuAT performs significantly better than existing single-step training methods" (L 272 - 273). However, based on the results in Table 1 and 2, the relative gain compared with GAT(-WA) is not significant. There are also no comparisons with GAT-WA in Table 2.


[A] Confidence-Calibrated Adversarial Training: Generalizing to Unseen Attacks; Stutz et al.; ICML 2020


[B] Adversarial examples improve image recognition; Xie et al.; CVPR 2020


---------------------------------------------------------------------------------------------------------------------------------------------------------------


Update after the rebuttal: I read the rebuttal and other reviews. The reviewers have addressed most of my concerns. I believe the paper meets the acceptance criteria for the conference. I vote for acceptance.


----------------------------------------------------------------------------------------------------------------------------------------------------------------

**Time Spent Reviewing:**

10

---

> ### Author Response · Authors · 2021-08-10
> **Response to reviewer 5nmX**
>
> We sincerely thank the reviewers for their time and valuable feedback on our work. We are happy to see that the reviewers find our work novel and significant, our evaluations comprehensive and thorough, and our results noteworthy in terms of effectiveness and efficiency. It is indeed motivating to find that all reviewers find our work to be worthy of acceptance at NeurIPS. We will certainly work towards incorporating all the suggestions in our camera-ready version.
>
> We address the specific comments posted by the reviewer *5nmX* below:
>
>  -   **Combining the proposed defense with AdvProp**
>       -   A drop in clean accuracy is indeed a concern in adversarial training which is being overcome in recent works by training on additional (synthetic) data [Ref: RobustBench leaderboard]. We believe such gains can be achieved by training our method using additional data as well.
>       -   The work AdvProp [B] aims to improve the clean accuracy of networks by training on weak adversarial examples with an auxiliary batch-norm layer. While the resulting network generalizes better to the ImageNet test set, and also to distortions of the ImageNet dataset, it is not effective against adversarial attacks. This is because, during test time, the batch-norm layer corresponding to clean samples is used. While the use of the auxiliary batch-norm layer can potentially lead to gains in robustness, it is unknown a priori whether the sample at test time is natural or adversarial, to decide on which batch norm layer is to be used.
>
>  -   **Comparison with AWP and GAT-WA in Table-2**
>
>      We did not include GAT-WA in the SOTA comparison table (Table-2) since this is not a method originally proposed by the authors of GAT [25]. We include results of GAT-WA in Tables-1(a) and 1(b) only to highlight that NuAT-WA outperforms baselines with weight-averaging as well. We understand GAT-WA can be a useful baseline for future work as well. We will certainly include these results (AWP and GAT-WA) for ImageNet-100 and MNIST as well in the final version of the paper.
>
> -   **Comparison with CCAT**
>
>     The method Confidence-Calibrated Adversarial Training [A] aims at rejecting adversarial examples, while the proposed method aims at improving the robustness to the same. The authors [A] however compare their work with TRADES [32] and PGD-AT [19] by adopting confidence-based thresholding to reject not more than 1% clean correct samples on these adversarial training baselines as well. As seen in Table-16 of the Supplementary in [A], on the CIFAR-10 dataset, RErr or robust error at $\tau=0$ (no rejection case) of CCAT against PGD-CE $\ell_\infty$ attack at $\varepsilon=0.03$ is 100%, showing that the method can only be used for detection of adversarial samples, and is not a suitable baseline for methods that improve robustness of models.
>
>
> -   **Gains of the proposed approach**
>   We report the gains of the proposed approach (NuAT and NuAT-WA) with respect to GAT and GAT-WA in the below table from the results reported in Table-2. We will update the paper to report the exact value of gain obtained in each case, rather than describing it as significant.
>
>      | Methods being compared                    | Dataset  | Clean (% gain) | AA (% gain) |
> |--------------------|----------|:--------------:|:-----------:|
> | NuAT(WA)  -  GAT(WA) | CIFAR-10 |      2.74      |     2.47    |
> | NuAT - GAT         | CIFAR-10 |      0.52      |     1.94    |
> | NuAT - GAT         | IN-100    |      1.02      |     3.04    |
> | NuAT - GAT         | MNIST    |      0.00      |     2.49    |

---

### Official Review · Reviewer_1Phm · 2021-07-16

**Rating:** 7
**Confidence:** 3

**Summary:**

This paper bridge the performance between single-step adversarial training and multiple-step adversarial training. The author proposed a Nuclear Norm regularizer in adversarial training (NuAT) which enhancing optimization to use joint batch-statistics of adversarial samples. Moreover, the author also utilizes the weight averaging method to boost the current performance. This work shows outstanding robust performance with significantly low computational cost.

**Limitations And Societal Impact:**

Authors did not clearly describe the limitation nor societal impact in the main manuscript in section 5,6,7.


**Main Review:**

This paper shows outstanding performance with single-step adversarial training. The experiment was thorough and easy to understand the paper. I think this paper has large significance in fast adversarial training research area.

However, I have few questions to be clarify about NuAT.
- I am curious about the nuclear norm itself. Does matching singular values is effective or matching clean and adversarial examples is effective? What if using KL divergence [TRADES] term instead of Nuclear norm? Does KL divergence does not show any benefits compare to the Nuclear norm?
- Does FGSM (Free is better than fast) + WA also work? How does performance change?
- Why does the Nuclear norm make a more diverse attack than the Frobenius norm? Any explanation?
- Why does the Nuclear norm able to make smoothed loss surface while the Frobenius norm can't?
- Does NuAT also work in ImageNet1K? How does performance shows in ImageNet 1K, not ImageNet100? Since fast adversarial learning is especially essential in the large-scale dataset, I think the results of ImageNet1K will make NuAT a more solid contribution in this research area.
- What is the difference between Nu-AT and NuAT using only CE+NN attack in supplementary table 4?
- What will be the performance when using NuAT using only NN attack?

Minor typo

Line 159: reference 29→27

**Time Spent Reviewing:**

12

---

> ### Author Response · Authors · 2021-08-10
> **Response to reviewer 1Phm**
>
> We sincerely thank the reviewers for their time and valuable feedback on our work. We are happy to see that the reviewers find our work novel and significant, our evaluations comprehensive and thorough, and our results noteworthy in terms of effectiveness and efficiency. It is indeed motivating to find that all reviewers find our work to be worthy of acceptance at NeurIPS. We will certainly work towards incorporating all the suggestions in our camera-ready version.
>
> We address the specific comments posted by the reviewer *1Phm* below:
>
>  -  **Matching singular values vs. matching clean and adversarial examples**
>
>     We would like to clarify that the proposed defense aims to minimize the Nuclear Norm of difference in pre-softmax values of clean and adversarial samples (Eq.2, L218-219). We do not match singular values of matrices comprising of clean and adversarial images respectively, which can possibly miss out on the one-to-one correspondence between images in the two matrices. Our method indeed aims to match the outputs of clean and adversarial samples similar to the KL divergence term in TRADES [32]. We expand on the difference w.r.t. the same in the following point.
>
>  -  **Use of KL divergence instead of Nuclear Norm**
>      -  While KL divergence can only be imposed on the softmax outputs of a classifier, the proposed regularizer can be imposed on the pre-softmax values as well, making it applicable to a wide range of regression tasks as well.
>      -   Even for the classification task, matching pre-softmax values results in better matching of representations between a clean and adversarial image leading to smoother representations. For example, let $f(x)$ represent the pre-softmax outputs of a neural network, and let $S(f(x))$ represent the Softmax predictions. Due to the shift-invariance of the Softmax activation function, we observe that $S(f(x)) = S(f(x)+g(x))$ where $g(x)$ is an arbitrary scalar function. This can potentially be a source of gradient obfuscation in single-step adversarial training, if $g$ is locally non-smooth. This would result in the generation of weaker single-step adversaries for training, leading to a drop in robustness. However, with the application of Nuclear Norm in the pre-softmax space, sufficient supervision can be provided to the network to distinguish between these two cases, leading to a further boost in robust performance.
>      -   Instance-specific losses such a Frobenius norm (GAT) and KL-divergence (TRADES) lead to a uniformly smooth representation (L161-162) while the proposed Nuclear-Norm regularizer uses batch-statistics to achieve function-smoothing only in the required dimensions and to the required extent (L167-168) leading to a better robustness-accuracy trade-off.
>      -   We performed an experiment by setting the number of attack steps to 1 in the official TRADES code. We obtain a robust accuracy of 38.36% which is significantly lower than our results (49.46%). We note that the authors also mention the same in Appendix D1 of their paper [32]: "*More specifically, we find that using one-step adversarial perturbation method like FGSM in the regularization term, defined in [KGB17], cannot defend against FGSM-k (white-box) attack.*"
>      -   Finally, to understand the impact of using KL divergence term instead of the Nuclear Norm regularizer in the proposed defense, we directly replace the latter with the KL divergence term in an ablation run. Using this, we obtain robust accuracy of 47.23% (against GAMA attack) after careful fine-tuning. We observe that increasing the value of $\lambda$ further leads to gradient masking. We therefore require the nuclear norm regularizer to achieve the additional gains in robust accuracy (49.46%). This experiment also highlights the importance of additional aspects of the proposed defense, such as the Bernoulli noise added initially, the linearly increasing $\lambda$ value, alternation of CE and CE+ NN attack, and cyclic LR schedule, which help in stabilizing single-step defenses, increasing the robustness with KL-divergence term from 38.36% to 47.23%.
>      -    We will include this discussion and the results obtained in the final version of our paper.
>  -  **FBF + WA**
>
>      We perform the experiment of adding weight-averaging (WA) to the official code of FBF [29]. Using WA, the clean accuracy degrades from 84.11% to 80.35% while the robust accuracy improves from 43.16% to 46.33%. Results without WA are slightly different from those reported in Table-1 as we had to rerun this experiment to report results with WA. While the gains in robust accuracy are significant on FBF as well, the final robust accuracy of the proposed method NuAT-WA is significantly higher (50.97%).
>  -  **Diversity of Nuclear Norm attack when compared to Frobenius Norm attack**
>
>      The nuclear norm of a matrix is the sum of its singular values ($\sum_i{\sigma_i}$), while the Frobenius norm is the $\ell_2$ norm of its singular values ($\sqrt{\sum_i{\sigma_i^2}}$). The $\ell_1$ norm of singular values  (nuclear norm) is related to the rank of the matrix. Minimization of $\ell_1$ norm leads to sparsity in singular values, resulting in a low rank matrix.
>
>      A Frobenius norm based attack tends to maximize the difference between clean and adversarial outputs of all samples in a batch individually, while the Nuclear Norm based attack additionally attempts to make the rows of the matrix independent in order to maximize the rank of the matrix. If we consider an example of a batch of 100 images belonging to the same class, Frobenius norm attack may result in all outputs going to the same class, while the Nuclear norm attack would push the outputs to diverse classes in order to reduce the dependence among different rows of the matrix. This can be observed from the confusion matrices in Fig.2 as well.
>
>  -  **Smoothness of loss surface when compared to Frobenius Norm**
>
>       Frobenius norm based defense also enforces smoothness of loss surface. The $\ell_2$ regularizer introduced in GAT [25] attempts to uniformly limit the oscillation of the network outputs along all dimensions, while the nuclear norm regularizer utilizes batch-statistics for achieving function smoothing only in the required dimensions, and to the required extent, leading to better robustness at the same clean accuracy, or in other words, a better robustness-accuracy trade-off.
>
>       Additionally, as discussed above, the nuclear-norm based attack is more diverse and stronger than the frobenius-norm based attack, which helps in achieving better robustness.
>
>  -  **Results on ImageNet1k**
>
>     Since the training of even a standard model (without adversarial training) on ImageNet1k requires large GPU memory and high computational requirements, we show the scalability of our approach by testing it on a 100-class ImageNet subset which can be trained at a tenth of the computational cost. We therefore demonstrate the scalability of the proposed method from 10 to 100 classes, from 32x32 to 224x224 image resolution and from small (ResNet18) to large (WideResNet-34-10) capacity models. Since we use a batch size of 64, we felt that if the proposed method is effective on a dataset with 100 classes, it would scale to 1000 classes as well. Even from past works [FBF, 29] we note that scalability of single-step defenses to large capacity models (WideResNet-34-10) has been a bigger challenge than scaling to larger datasets. As shown in Table-1(a,b), FBF does not improve results from ResNet-18 to WideResNet-34-10, while the proposed method NuAT (even without weight averaging) achieves 1.57% higher robust accuracy and 4.29% higher clean accuracy on WideResNet-34-10. From Table-6 in the Supplementary, we also note that the proposed approach NuAT achieves 4.16% boost in clean accuracy and 3.86% boost in robust accuracy when the network is scaled from ResNet-18 to ResNet-34 on ImageNet-100.
>
>     However, we agree that showing results on ImageNet1k would be a very good contribution to the field. We will work towards this, and try our best to include the results in the final version.
>
>  -  **Difference between NuAT and NuAT using only CE+NN attack**
>
>      In the proposed defense we use R-FGSM (CE) attack and R-FGSM (CE) + Nuclear Norm (NN) attack in alternate training iterations to obtain strong and diverse attacks for training (L206-211 of the main paper). In the ablation A2 in Table-4 (NuAT using only CE+NN attack), we use only the R-FGSM (CE) + Nuclear Norm (NN) attack in all training iterations. This results in a small drop in robust accuracy at a higher clean accuracy, highlighting the importance of using R-FGSM (CE) attack in alternate training iterations to boost diversity, as was observed in prior work [GAT, 25] as well.
>
> -  **NuAT using only NN attack**
>
>      The Nuclear Norm regularizer acts as a smoothing term to overcome gradient masking and improve the diversity of single-step attacks. However it does not provide a strong misclassification objective such as the Cross-entropy loss for single-step training (L200-202). We obtain clean accuracy of 83.59% and robust accuracy (against GAMA PGD 100 attack [25]) of 45.56% using only NN attack for training. This improves to 82.23% clean, and 48.61% robust accuracy with the use of weight-averaging. We will include this ablation in the final version.
>
> -   **Limitations and societal impact**
>
>     We will include this section in the final version. The proposed defense does not provide certified guarantees on robustness. This could be overcome by incorporating randomized smoothing techniques along with the proposed defense [A]. Further, adversarial training typically results in lower accuracy on clean samples when compared to standard training.
>
>      [A] Salman et al., Provably Robust Deep Learning via Adversarially Trained Smoothed Classifiers, NeurIPS 2019

---

> > ### Comment · Reviewer_1Phm · 2021-08-25
> > **Thank you for your response**
> >
> > Thank you for detail response.
> >
> > I think most of my concerns are resolved.
> > If author can provide the ImageNet1K results until the discussion period ends it will be much better.
> >
> > I will keep my score.

---

> > > ### Author Response · Authors · 2021-08-29
> > > **Results on ImageNet-1k**
> > >
> > > We sincerely thank all reviewers for their time in going through our response and for the post-rebuttal comments.
> > >
> > > We integrated our code with the ImageNet-1k training code from the FBF [29] repository, since this has several optimizations such as half-precision computations and training in 3 phases on images with increasing image resolutions while adjusting the batch size accordingly. We report results of the FBF baseline and NuAT (Ours) on images resized to $352 \times 352$ and cropped to $288 \times 288$ in the below table. Similar to FBF, we consider robustness within the $\ell_\infty$ constraint of $4/255$. We use our base single-step training (NuAT) and expect further improvements using NuAT-H, NuAT2 and NuAT2-WA.
> > >
> > > |                | Clean Accuracy (%) | PGD (50-step) (%) | GAMA-PGD  (100-step) (%) |
> > > |----------------|:--------------:|:-------------:|:--------------------:|
> > > |     FBF    |      56.70     |     31.96     |         27.44        |
> > > |   NuAT (Ours)  |      58.65     |     34.28     |         29.86        |
> > > | NuAT-WA (Ours) |      59.27	          |          34.97	     |       30.52               |
> > >
> > > While the results reported in the FBF paper are based on evaluations on PGD 50-step attack, we additionally report results against GAMA PGD-100 attack to get a more reliable estimate of the true robustness. Using NuAT, we achieve 1.95% boost in clean accuracy and 2.42% boost in robustness against the GAMA PGD 100-step attack. By using Weight averaging (NuAT-WA), we obtain a further boost in both clean accuracy and robustness.
> > >
> > > This demonstrates that the proposed approach indeed scales to ImageNet-1k and achieves a boost over FBF as well.
> > >
> > > An important contribution of our work is in being able to run longer training schedules and achieve results that are better than multi-step training methods such as PGD-AT and TRADES. Since even standard ImageNet-1k training on 100-epoch long schedules is computationally intensive, we are unable to report these results currently.

---

> > > > ### Comment · Reviewer_1Phm · 2021-08-31
> > > > **Thank you for your response**
> > > >
> > > > I think this submission is publication-worthy in NeurIPS 2021.
> > > > I will keep my score.
> > > >
> > > > Thanks again to the author for the extensive response during the rebuttal.

---

### Decision · Program_Chairs · 2021-09-27

**Decision:**

Accept (Poster)

**Comment:**

The paper introduces a Nuclear-Norm Adversarial Training (NuAT) by imposing a rank minimization constraint on the oscillation of function values across a training minibatch. All reviewers thought the paper is above the accept threshold. I suggest authors to take reviewers' suggestions in revising the final version of their paper.